



# Assessing the feasibility of a directional CRNS-sensor for estimating soil moisture

Till Francke[1], Maik Heistermann[1], Markus Köhli[8,9], Christian Budach[1], Martin Schrön[2], and Sascha E. Oswald[1]

[1]Institute of Environmental Science and Geography, University of Potsdam, Karl-Liebknecht-Straße 24–25, 14476 Potsdam, Germany
[2]UFZ – Helmholtz Centre for Environmental Research GmbH, Dep. Monitoring and Exploration Technologies, Permoserstr. 15, 04318, Leipzig, Germany
[8]Physikalisches Institut, Heidelberg University, Im Neuenheimer Feld 226, 69120 Heidelberg, Germany
[9]Physikalisches Institut, University of Bonn, Nussallee 12, 53115 Bonn, Germany

**Correspondence:** Till Francke (francke@uni-potsdam.de)

**Abstract.** Cosmic Ray Neutron Sensing (CRNS) is a non-invasive tool for measuring hydrogen pools like soil moisture, snow, or vegetation. The intrinsic integration over a radial hectare-scale footprint is a clear advantage for averaging out small-scale heterogeneity, but on the other hand the data may become hard to interpret in complex terrain with patchy land use.

This study presents a directional shielding approach to block neutrons from certain directions and explores its potential to gain a sharper view on the surrounding soil moisture distribution.

Using the Mont-Carlo code URANOS, we modelled the effect of additional polyethylene shields on the horizontal field of view and assessed its impact on the epithermal count rate, propagated uncertainties, and aggregation time.

The results demonstrate that directional CRNS measurements are strongly dominated by isotropic neutron transport, which dilutes the signal of the targeted direction especially from the far field. For typical count rates of customary CRNS stations, directional shielding of halfspaces could not lead to acceptable precision at a daily time resolution. However, the mere statistical distinction of two rates should be feasible.

## 1 Introduction

### 1.1 Cosmic ray neutron sensing in environmental sciences

The use of CRNS in the environmental sciences has considerably increased in the past decade. Its main application is the measurement of soil water content (Zreda et al., 2008). Such measurement could serve a variety of purposes in both, research and application, for example to close the water balance in atmospheric or hydrological models (Schreiner-McGraw et al., 2016; Dimitrova-Petrova et al., 2020), or to support irrigation management (Li et al., 2019; Franz et al., 2021) or snow cover analysis (Schattan et al., 2017). Furthermore, the use of CRNS to independently estimate biomass or even interception by vegetation has showed potential (Baroni and Oswald, 2015; Baatz et al., 2015).





20    A major advantage of the CRNS method is its non-invasive character, as opposed to traditional measurements of soil moisture. Additionally, the measured cosmic-ray neutrons naturally integrate over an area with approximately 150 m radius and a depth of decimeters (Köhli et al., 2015), which results in practical and representative estimates of soil moisture at the field scale. This intermediate scale of measurement support effectively bridges the gap between traditional point measurements and coarser large-scale products from remote sensing or hydrological modelling.

However, the larger spatial support of the omni-directional measurement comes at the cost of spatial resolution. It could introduce a crucial systematic bias especially at sites of highly non-homogeneous land use, such as patchy soil moisture distribution, snow cover patterns, or vegetation (Franz et al., 2013; Coopersmith et al., 2014; Lv et al., 2014; Schrön et al., 2017; Schattan et al., 2019).

Several strategies have emerged in the last years to address this challenge, such as areal correction (Schrön et al., 2018b), 30    energy level separation (Rasche et al., 2021), spatial inversion (Heistermann et al., 2021), or sequential measurements with mobile CRNS roving (Franz et al., 2015; Schrön et al., 2018a; Fersch et al., 2018; Vather et al., 2020; Zhang et al., 2021). The latter method can only produce campaign-based snapshots in time, while large detectors are needed to compensate the short integration interval. Heistermann et al. (2021) have reconstructed sub-footprint patterns of soil moisture using a dense, partially overlapping network of CRNS sensors. Although this approach provides spatially and temporally continuous data, it 35    requires a high number of instruments, and is based on strong assumptions on spatial continuity in terms of soil water content. Rasche et al. (2021) reported the concomitant use of epithermal and thermal neutron counts to exploit their different footprint characteristics. Combining these with process knowledge of diverse hydrological units in the footprint allowed to disentangle the sensor signal between the near and far field.

Finally, directional CRNS measurements could provide a way of altering the omni-directional footprint towards a target field 40    of view. This would not only open the routes towards a scanning mode for CRNS with a spatial or at least angular resolution within the footprint, but also towards blocking off undesired parts of a (stationary or mobile) footprint which would otherwise introduce bias (such as forests, lakes, or urban structures).

## 1.2    Existing directional neutron measurements

"Directional neutron measurements" refers to the measurement of a neutron flux coming from a specific direction. Such mea- 45    surements usually aim for localising a neutron source or even for pixelwise imaging of a far distant object or at an imaging facility. This can be achieved by focusing the emissions of the neutron source and/or by masking out neutrons arriving at the sensor from a certain direction. Evidently, controlling the neutron emission is only possible when artificial sources are used. In this case, high-energy neutrons, collimators and/and short distances allow for high spatial resolutions and imaging methods in medicine, material research (e.g. neutron radiography, Kardjilov et al., 2018) and fast neutron tomography (e.g. Tötzke et al., 50    2019), imaging of artifacts (e.g. Lehmann et al., 2010), defense, and homeland security (e.g. pulsed fast neutron analysis, Gozani, 1995), Hamel et al. (2017).

The CRNS method, however, relies on the natural uncontrolled cosmogenic neutron flux, lower energies, and longer distances between sensor and object (meters to hectometers). Hence, the above concepts to control neutron emission do not apply.





Instead, directional measurements could be obtained by a partial enclosure ("shielding" or "collimator") of the sensor with a

material that absorbs or slows down the incoming neutrons. Neutrons arriving from the shielded sides are therefore less likely to be counted by the detector.

In planetary sciences, the directional measurement of neutrons helped to map water distribution on Moon (Feldman et al., 1999) and Mars (Mitrofanov et al., 2018). These spaceborne directional neutron detector use a directional shielding (collimator) made of PE, allowing only neutrons from the "collimation field of view (FOV)" to enter the detector. The very thin atmosphere

on Mars, combined with the comparatively high energy level of the neutrons enabled a favourable "collimation efficiency"

$$\eta = \frac{N_{\mathrm{FOV}}}{N_{\mathrm{total}}} \tag{1}$$

being the ratio between counts from the FOV $N_{\mathrm{FOV}}$ and the total counts $N_{\mathrm{total}}$. This allowed mapping the Mars surface from approx. 400 km above the ground with a relatively high spatial resolution. For applications on Earth, however, such long-range measurements are unfeasible due to its much denser atmosphere. We will use this quantity $\eta$ as the *directional contribution*

being collected from the targeted field of view (FOV) as fraction of the total counts detected.

In environmental sciences, Zreda et al. (2020) have recently patented a downward-looking CRNS sensor. While this approach clearly aims at retrieving the signal from short distances (i.e. the area directly below the downward-looking sensor), it likewise involves directional shielding by blocking neutrons reaching the sensor from other directions.

In 2018, we constructed a directional shielding as an add-on for a commercially available CRNS-sensor, a CRS 2000B

(HydroInnova, see Fig. 1). Its purpose is to confine the measurement towards the direction of the area at the unshielded side (FOV) by reducing the contribution from the part of the footprint outside the FOV. The directional shielding was also designed to allow a stepwise turning of the partly-shielded detector by a controlled stepper motor, and thus stepwise change of the FOV. This could be operated to cover the full $2\pi$ periphery, in flexible angular sections, and thus allow for a scanning mode producing time-series of count rates for different FOV in the footprint. The extent and thickness of the directional shielding

were a compromise in order to keep size and weight in manageable proportions, and also its opening was large enough to obtain still count rates and integration times in the range of standard CRNS applications. The shielding consists of 4.5 cm of layered boron-loaded polyethylene at three sides, top and bottom of the detector to moderate incoming neutrons. The inside of this chamber is additionally coated with 5 mm boron carbide to absorb the remaining thermalized neutrons. This configuration is used in the presented study as an example case for the theoretical development and neutron scattering simulations. Systematic

practical tests will then be designed on the findings.

### 1.3 Uncertainty in CRNS-measurements

The neutron count rate $R$ [counts h$^{-1}$] registered by a detector is a function of the incoming neutron flux, detector sensitivity and hydrogen pools in its footprint. Assuming a stationary setting of these factors and disregarding all errors and uncertainties, this corresponds to an expected value of total counted neutrons $N$ [counts] in a measurement period of $\Delta t$ [h] of

$$N = R \cdot \Delta t. \tag{2}$$





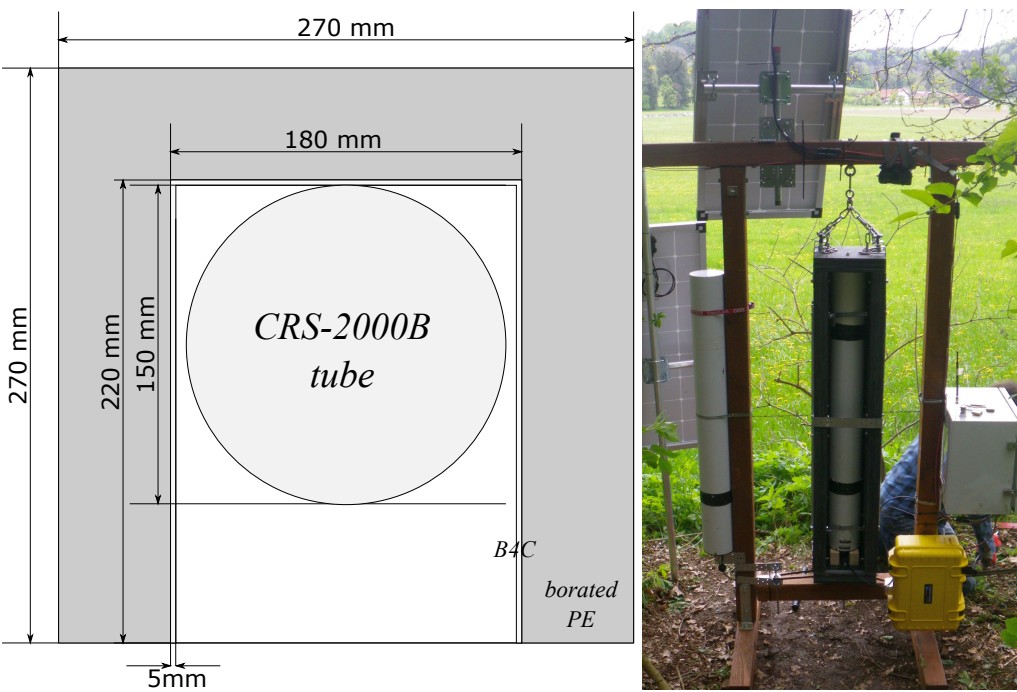

**Figure 1.** CRNS sensor prototype with directional shielding, operated on the boundary between forest and meadow. Left: horizontal cross section; right: Installation with automatic directional control. The white box contains the logger and modem, the yellow box hosts the electronic controls for a stepper motor controlling the direction of the sensor. A conventional counting tube is mounted to the left.

However, when dealing with real measured neutron counts $N$, two sources deteriorating the signal must be considered: (1) statistical error of counts and (2) device noise / intended counts.

(1) Due to the stochastic nature of the counting process, the number of neutrons $N$ (i.e. the realization of a measurement) is subject to a Poisson-type error $\sigma$. From the Poisson-characteristics of the counting process follows that

$$\sigma = \sqrt{N}, \tag{3}$$

with $\sigma$ being the standard deviation of all the values of $N$ we would get when repeating the measurement in the given interval.

(2) A typical CRNS-detector is targeted at counting epithermal neutrons ($0.5\,\mathrm{eV} < E < 10^5\,\mathrm{eV}$, (Köhli et al., 2018)), because these neutrons show sensitivity to the the abundance of hydrogen within the footprint. However, the counting is affected by additional untargeted neutron counts $N_{!epi}$, e.g. caused by ionizing particles from other energy levels and particles, terrestrial radiation and background radioactivity of detector materials (Weimar et al., 2020). $N_{!epi}$ is associated with an error of $\sigma_{!epi}$. Thus, the overall uncertainty $\sigma_{\mathrm{total}}$ when measuring $N_{\mathrm{total}}$ is a superposition of the two error sources.

The above-mentioned errors can be characterized by appropriate distributions and their respective parameters:

Concerning neutrons counted of other energy levels $N_{!epi}$, the measured signal contains a fraction $\epsilon$ of such "bycatch" flux. For a typical gas detector, $\epsilon$ results from approx. 20 % thermal and 10 % counts from fast neutrons (Baatz et al., 2015; Köhli





et al., 2021) in the measured signal (the precise values depend on ambient hydrogen pools and chosen cutoff thresholds). Consequently, the actual number of counted neutrons $N_{\text{total}}$ is

$$N_{\text{total}} = N_{\text{epi}} + N_{\text{!epi}} = N_{\text{epi}} + \epsilon \cdot N_{\text{total}} = \frac{1}{1 - \epsilon} N_{\text{epi}}, \qquad (4)$$

where $\epsilon$ is the fraction of bycatch neutrons.

Analogously to Eq. (3), the respective additional noise introduced with $N_{\text{!epi}}$ is

105 $$\sigma_{\text{!epi}} = \sqrt{N_{\text{!epi}}} = \sqrt{\frac{\epsilon}{1 - \epsilon} \cdot N_{\text{epi}}} \qquad (5)$$

According to our measurements, counts caused by radioisotope contamination of the detector walls are in the order of 0.5 counts h$^{-1}$ per m$^2$ of the proportional counter (cf. Dębicki et al. (2011) for comparable tubes). This amounts to approx. 0.5 counts h$^{-1}$ for a standard CRS1000. Like other counts of unintended ionizing radiation (detector gas, protons and muons, etc.) these can be effectively filtered out by appropriate detector threshold settings (Quaesta Instruments, pers. communication), and are thus

ignored here. We also consider temperature effects (reported for neutron monitors, e.g. by Krüger et al., 2008) as negligible, as we are dealing with a pairwise concomitant operation (see Fig. 3), in which such effects would be cancelled out. Noise from electronic components (e.g. from external fields) is in the order of $< 10$ counts h$^{-1}$ for typical detectors, and can be effectively eliminated with appropriate threshold settings (Quaesta Instruments, pers. communication). Thus, it is also ignored here.

### 1.4 Specific challenges with directional neutron measurements in CRNS

Generally, CRNS-measurements are affected by the uncertainties described in section 1.3. For directional CRNS-measurements, additional challenges arise:

– Incoming cosmogenic fast neutrons are converted to epithermal neutrons by hydrogen pools within the footprint. In this context, we denote the location of this conversion as 'origin' as it constitutes the place for which we infer the information when measuring the neutron. However, most of these epithermal neutrons do not reach the sensor directly.

Instead, neutrons arriving at the sensor have usually experienced multiple elastic collisions, resulting in an irregular trajectory. Consequently, the incidence angle of a neutron reaching the detector is only loosely correlated with its angle of origin. Thus, a directional shielding can only imperfectly filter epithermal neutrons for this direction of origin.

– A directional shielding blocks neutrons reaching the detector from certain angles. This blockage can be achieved by a sufficiently thick blocking material, e.g. layers of High-density polyethylene (HDPE). For practical reasons, however,

compromises between blocking properties and size/weight constraints have to be made. Thus, the directional blocking will be imperfect, allowing a certain fraction of neutrons to reach the detector also from the shielded side.

– Blocking neutrons arriving at the detector from certain angles effectively reduces the overall count rate. Consequently, longer integration times are required for obtaining the same number of counts in a given environment.





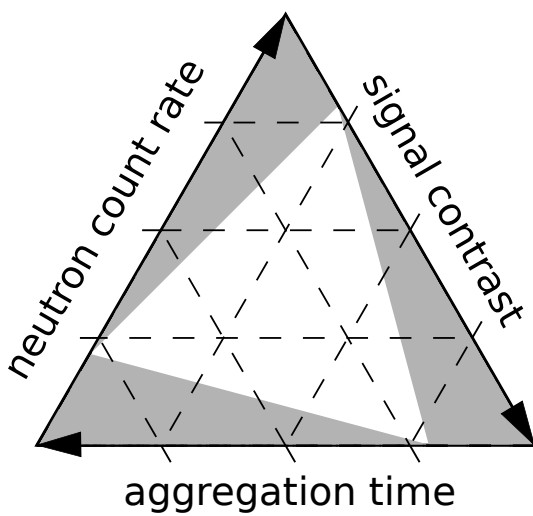

**Figure 2.** "Tradeoff-triangle" in directional CRNS-measurements as ternary plot: for a given combination of two of these parameters, the third parameter must be adjusted accordingly to obtain the requested accuracy. The inner triangle illustrates the parameter space for an increase in required accuracy. The straight outline here is only chosen for the sake of simplicity and could be curved instead.

Directional neutron sensing aims to determine the neutron flux rate $R_1$ that is characteristic of the area $A_1$ in the field of view. The flux rate $R_2$ of the complementary area $A_2$ may also be of interest in itself, or effectively only be a factor influencing the measurement. The signal contrast between the two ($\Delta R$, see Eq. (19)) is eventually determined by the different hydrogen inventories in the two areas.

Summarizing the general uncertainties (section 1.3) and the limitations specific to directional measurements, we may conceptualize the determination of these pools via directional neutron measurement of $R_1$ (and $R_2$) as a trade-off problem with the parameters count rate ($R_1$), the signal contrast and the aggregation time ($\Delta t$): To obtain measurements for $R_1$ at a given accuracy, two of these parameters (e.g. count rate and signal contrast) determine the minimum of the third parameter (e.g. aggregation time, see Fig. 2). If we raise the required accuracy, the lower limits of the three parameters need to be increased (grayed-out areas in Fig. 2). The same applies if the relative noise of the measurement increases (e.g. by a higher device noise or a narrower shielding angle).

Systematically exploring these dependencies and limitations is a prerequisite for the design and application of directional CRNS measurements. However, in situ experiments are impracticable, as they require well-defined setups with spatially-known neutron flux rates and multiple configurations of sensors and shieldings. Instead, we propose to use simulations of neutron scattering and neutron detection in order to better understand and quantify potentials and limitations of directional CRNS measurements. Our specific objectives are as follows:

1. Quantify directional specificity and reduction in neutron count of a directional CRNS-sensor setup





2. Assess the informative value of directional CRNS measurements as a function of signal aggregation time, count rates, signal contrast and sensor noise, and the respective limits of applicability

Objective 2 can be addressed from different perspectives:

A: What temporal resolution $\Delta t$ could be obtained?

B: What spatial contrast in the count rates $\Delta R$ can be resolved?

C: What count rates $R$ are required to yield robust estimates?

Each of these points (A-C) can be addressed when looking at either (I) *determining* count rates at a chosen accuracy or (II) statistically *distinguishing* two count rates. These are somewhat different objectives with different requirements, which will be further demonstrated in the results section.

## 2 Methods and data

By applying the neutron simulation model URANOS (section 2.1.1), we assess the characteristics of angular shielding (section 2.1.2). As these simulations are computationally demanding, we then generalize the findings in an analytical approach (section 2.2) in order to outline the possible range of application of directional CRNS measurements.

### 2.1 Neutron simulation

#### 2.1.1 Neutron simulation model URANOS

The Monte-Carlo tool URANOS (Köhli et al., 2015) is designed specifically for modeling neutron interactions within the environment in the framework of CRNS. The standard calculation routine features a ray-casting algorithm for a single neutron propagation and a voxel engine. Instead of propagating particle showers in atmospheric cascades, URANOS reduces the computational effort and makes use of the analytically defined cosmic-ray neutron spectrum by Sato (2015). The URANOS model uses setups with either open domains of at least 600 m or smaller sizes with periodic or reflecting boundary conditions.

#### 2.1.2 Scenarios simulated with URANOS

The simulation geometry consists of a ground layer with a thickness of 1.3 m and a 1000 m air (buffer) layer. The soil consists of 50 %$_{\text{Vol}}$ solids and a scalable amount of $H_2O$. The solids are composed of 75 %$_{\text{Vol}}$ $SiO_2$ and 25 %$_{\text{Vol}}$ $Al_2O_3$ at a compound density of 2.86 g cm$^{-3}$. The air medium consists of 78 %$_{\text{Vol}}$ nitrogen, 21 %$_{\text{Vol}}$ oxygen and 1 %$_{\text{Vol}}$ argon at a pressure of 1020 mbar. The air humidity is set to 7 g m$^{-3}$ and soil moisture is set to 10 %$_{\text{Vol}}$. In this study three different simulation setups have been used:

1. In order to assess the effect of adding a further moderator (i.e. directional shielding) on the measured intensity, in a first setup, the detector with its actual dimensions has been placed inside a domain of 10 m lateral extension with periodic boundaries and a cosmic neutron source.





**Table 1.** Symbols used in the text

| Symbol | Explanation |
|---|---|
| A | matrix holding coefficients for mixing of albedo neutrons |
| B | matrix holding coefficients for mixing of all neutrons |
| $\beta$ | count rate reduction factor due to shielding |
| $\Delta R$ | relative difference in the count rates $R_1$ and $R_2$ |
| $\Delta t$ | aggregation interval |
| $\epsilon$ | fraction of non-epithermal neutrons |
| $\eta$ | directional contribution of shielding |
| $\gamma$ | fraction of albedo neutrons in epithermal neutrons |
| J | matrix of ones |
| N | number of counted neutrons |
| $p_{thresh}$ | threshold p-value |
| R | count rate |
| $\sigma$ | error expressed as standard deviation |
| $\theta$ | volumetric soil moisture |
| *sub-/superscripts* | |
| ! | non-.... |
| 05 | concerning statistical distincion of rates |
| albedo | albedo neutrons, having interacted with hydrogen pools |
| determ | concerning determination of rates |
| dir | directional |
| epi | epithermal |
| f | shielded, i.e. directional detector facing towards a half-plane |
| no | no shield, unshielded detector |
| r | reconstructed from directional measurement |
| s | shielded, i.e. directional detector |
| total | overall counts |



2. Secondly, the to-scale-model of the detector has been used to calculate the response function (Köhli et al., 2018) of each face of the detector by probing the response to a series of monoenergetic neutrons released perpendicular to its face.

3. Thirdly, a virtual detector has been placed in a large-size domain. This entity was equipped with the beforehand found response functions and had, in order to collect enough statistics, a geometry which is slightly larger than the physical instrument. For the purpose of investigating the distance dependent angular response it can be regarded as a suitable approximation. This spherical virtual detector has been placed at a height of $1.75\,\mathrm{m}$ with a radius of $1.25\,\mathrm{m}$ within a domain size of $800\,\mathrm{m} \times 800\,\mathrm{m}$. Cosmic neutrons are released at a height of $50\,\mathrm{m}$ using a circular source with a radius of $400\,\mathrm{m}$. Thermal neutron transport was disabled for reasons of computational speed. This configuration leads to a homogeneous neutron flux distribution within the innermost $400\,\mathrm{m} \times 400\,\mathrm{m}$. Origin data can be retrieved for a radius of approximately $300\,\mathrm{m}$.

## 2.2 Generalisation of interpreting directional measurements

Aiming to generalize the the findings of the neutron simulations (sections 3.2 and 3.3), we conceptualize the directional measurement as a linear mixing and analytically express the corresponding propagation of errors for an simplistic geometric setup.

### 2.2.1 Description of idealized geometric setup

For the sake of simplicity, we use the simplest geometric setup imaginable for directional CRNS-measurements: A plain divided into two homogeneous half-spaces $A_1$ and $A_2$. Each half-space corresponds to a count rate $R_1$ and $R_2$, respectively, when measured far enough from its borders. At the border of both half-spaces, we implement a directional detector. When facing $A_1$, it yields the count rate $R_{\mathrm{f1}}$; when directed towards $A_2$, it registers the count rate $R_{\mathrm{f2}}$. Both count rates are measured and used in the computations. Consequently, the evaluation employ a FOV of $180°$ ($\pi$).

### 2.2.2 Components of the CRNS-signal

The total count rate $R_{\mathrm{total}}$ registered by a sensor results from non-epithermal and epithermal neutrons (see Eq. (4)).

The epithermal counts, in turn, are composed of counts from albedo neutrons having interacted with the hydrogen pools of interest ($R_{\mathrm{alb}}$) and non-albedo neutrons ($R_{\mathrm{non\text{-}alb}}$) without such interaction. We denote the respective percentage as $\gamma$ (see Fig. 4, left bar), so

$$R_{\mathrm{alb}} = (1 - \gamma) \cdot R_{\mathrm{epi}}. \tag{6}$$

Please note that $\gamma$ differs between the omni-directional and the directional detector, which will be shown later. Therefore we distinguish in the following between a $\gamma_{\mathrm{s}}$ and a $\gamma_{\mathrm{no}}$ for the directional and the omni-directional detector, respectively.





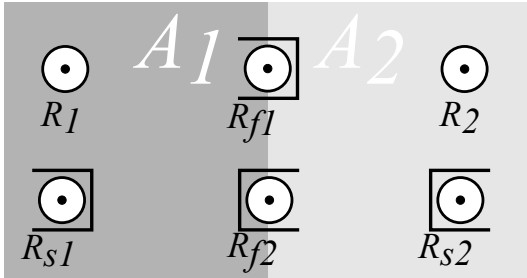

⊙ detector
◉ directional detector

**Figure 3.** Geometric setup of experiment: $R_1$ and $R_2$ denote the CRNS count rates within the half-planes $A_1$ and $A_2$, respectively. $R_{s1}$ and $R_{s2}$ symbolize the operation with directional shielding (creating a "directional CRNS sensor"). $R_{f1}$ and $R_{f2}$ denote the count rates the directional sensor registers when placed on the border and pointed towards $A_1$ or $A_2$, respectively.

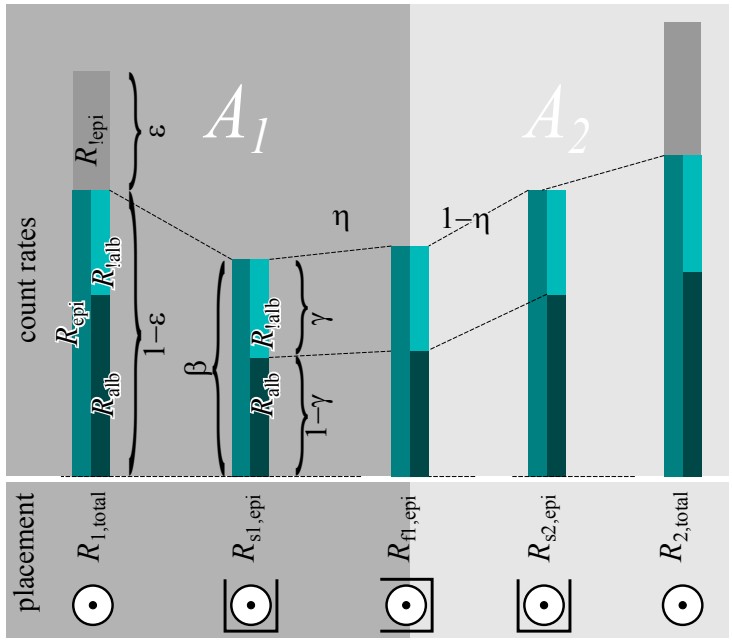

⊙ CRNS detector
◉ directional CRNS detector

**Figure 4.** Components of a CRNS signal for omni-directional and directional detectors. Left bar: partitioning of the total count rate ($R_{\text{total}}$) into spurious $R_{\text{non-epi}}$ and counts of epithermal neutrons ($R_{\text{epi}}$), in turn resulting from albedo ($R_{\text{alb}}$) and non-albedo ($R_{\text{non-alb}}$) neutrons. Second bar: reduced count rate $R_s$ and increase of no-albedo fraction $\gamma$ by adding the directional shielding; third bar: mixing of signal with directional detector placed between the half-planes $A_1$ and $A_2$.





For the directional sensor placed entirely into the interior of one of the half-planes, $A_1$ or $A_2$, the directional shielding causes partial blockage of the neutrons arriving, reducing its count rate by the factor $\beta$ to $R_s$ (see Fig. 4, second bar):

$$R_{\text{s,epi}} = \beta \cdot R_{\text{epi}}, \tag{7}$$

Values for $\gamma$ and $\beta$ were obtained from neutron simulations as discussed in section 2.1.2, setup 1.

### 2.2.3 Mixing and un-mixing of signals from directional sensors

When placing the directional sensor exactly between the two half-planes $A_1$ and $A_2$, the components of its counts (albedo and non-albedo) can be described as a mixing from the adjacent planes. The albedo neutrons mix according to the orientation of the shielding and the respective directional contribution $\eta$ (Eq. (1)), i.e. ratio between counts from the desired angle and total counts (see Fig. 4, third bar):

$$R_{\text{f1,alb}} = \eta \cdot R_{\text{s1,alb}} + (1 - \eta) \cdot R_{\text{s2,alb}} \tag{8a}$$

$$R_{\text{f2,alb}} = \eta \cdot R_{\text{s2,alb}} + (1 - \eta) \cdot R_{\text{s1,alb}}. \tag{8b}$$

While the directional detector may be oriented towards the left half plane $A_1$ or the right half plane $A_2$, there is no fundamental difference between the two and we exemplify the next step with the one oriented towards $A_1$ and its count rate $R_{\text{f1,alb}}$.

Substituting Eq. (8a) with the Eqs. (6), (7) and then (4) yields

$$
\begin{aligned}
R_{\text{f1,alb}} &= (1 - \gamma_{\text{s}}) && (\eta \cdot R_{\text{s1,epi}} + (1 - \eta) \cdot R_{\text{s2,epi}}) \\
&= (1 - \gamma_{\text{s}}) \cdot \beta && (\eta \cdot R_{\text{1,epi}} + (1 - \eta) \cdot R_{\text{2,epi}}) \\
&= (1 - \gamma_{\text{s}}) \cdot \beta \cdot (1 - \epsilon) && (\eta \cdot R_{\text{1,total}} + (1 - \eta) \cdot R_{\text{2,total}}).
\end{aligned}
\tag{9}
$$

Considering both rates $R_{\text{f1,alb}}$ and $R_{\text{f2,alb}}$ simultaneously in a vector $\boldsymbol{R}_{\text{f,alb}}$, we may write in matrix notation:

$$\boldsymbol{R}_{\text{f,alb}} = \begin{pmatrix} R_{\text{f1,alb}} \\ R_{\text{f2,alb}} \end{pmatrix} = (1 - \gamma_{\text{s}}) \cdot \beta \cdot (1 - \epsilon) \begin{pmatrix} \eta & 1 - \eta \\ 1 - \eta & \eta \end{pmatrix} \begin{pmatrix} R_{\text{1,total}} \\ R_{\text{2,total}} \end{pmatrix} = (1 - \gamma_{\text{s}}) \cdot \beta \cdot (1 - \epsilon) \cdot \mathbf{A} \cdot \boldsymbol{R}_{\text{total}}, \tag{10}$$

where $\mathbf{A}$ is a symmetric matrix containing the coefficients for the mixing. For the non-albedo neutrons $\boldsymbol{R}_{\text{f,non-alb}}$, the mixing is simply the average of the corresponding rates

$$\boldsymbol{R}_{\text{f,!alb}} = \frac{1}{2} \cdot \mathbf{J} \cdot \boldsymbol{R}_{\text{s,!alb}} = \frac{1}{2} \cdot \mathbf{J} \cdot \gamma_{\text{s}} \boldsymbol{R}_{\text{s,epi}} = \frac{1}{2} \cdot \mathbf{J} \cdot \gamma_{\text{s}} \cdot \beta \boldsymbol{R}_{\text{epi}} = \frac{1}{2} \cdot \mathbf{J} \cdot \gamma_{\text{s}} \cdot \beta \cdot (1 - \epsilon) \boldsymbol{R}_{\text{total}}, \tag{11}$$

where $\mathbf{J}$ is the matrix of ones.

Applying Eq. (4) for the non-epithermal counts leads to

$$\boldsymbol{R}_{\text{f,!epi}} = \frac{\epsilon}{1 - \epsilon} \cdot \boldsymbol{R}_{\text{f,epi}} = \frac{\epsilon}{1 - \epsilon} \cdot (\boldsymbol{R}_{\text{f,alb}} + \boldsymbol{R}_{\text{f,!alb}}) = \epsilon \cdot \beta \cdot \left( (1 - \gamma_{\text{s}}) \cdot \mathbf{A} + \frac{1}{2} \cdot \mathbf{J} \cdot \gamma_{\text{s}} \right) \boldsymbol{R}_{\text{total}}. \tag{12}$$



Adding Eqs. (10), (11), and (12) yields the total count rates of the directional detector $\boldsymbol{R}_{\mathrm{f,total}}$

$$
\begin{aligned}
\boldsymbol{R}_{\mathrm{f,total}} &= \boldsymbol{R}_{\mathrm{f,alb}} + \boldsymbol{R}_{\mathrm{f,!alb}} + \boldsymbol{R}_{\mathrm{f,!epi}} \\
&= (1-\epsilon)\cdot\beta\cdot\left((1-\gamma_{\mathrm{s}})\cdot\mathbf{A} + \frac{1}{2}\cdot\mathbf{J}\cdot\gamma_{\mathrm{s}} + \frac{\epsilon}{1-\epsilon}\left((1-\gamma_{\mathrm{s}})\cdot\mathbf{A} + \frac{1}{2}\cdot\mathbf{J}\cdot\gamma_{\mathrm{s}}\right)\right)\boldsymbol{R}_{\mathrm{total}} \\
&= \left(\mathbf{A}(\beta-\beta\gamma_{\mathrm{s}}) + \frac{1}{2}\beta\gamma_{\mathrm{s}}\mathbf{J}\right)\boldsymbol{R}_{\mathrm{total}} = \mathbf{B}\boldsymbol{R}_{\mathrm{total}}
\end{aligned} \qquad , \qquad (13)
$$

with $\mathbf{B}$ being a matrix summarizing all the operations.

For reconstructing the unshielded count rates in the half-spaces ($R_{\mathrm{total}}$) from the two directional measurements ($R_{\mathrm{f1}}$ and $R_{\mathrm{f2}}$), we use the inverse operation:

$$
\boldsymbol{R}_{\mathrm{r,total}} = \mathbf{B}^{-1}\boldsymbol{R}_{\mathrm{f}} = \frac{1}{k_1}\begin{pmatrix} k_2 & k_2+2 \\ k_2+2 & k_2 \end{pmatrix}\boldsymbol{R}_{\mathrm{f,total}}, \qquad (14)
$$

where

$$
k_1 = 2\beta(2\gamma_{\mathrm{s}}\eta - \gamma_{\mathrm{s}} - 2\eta + 1), \qquad\qquad k_2 = (2\gamma_{\mathrm{s}}\eta - \gamma_{\mathrm{s}} - 2\eta). \qquad (15)
$$

The subscript "$r$" in Eq. (14) denotes the *reconstructed* rates for estimating the true (and unknown) ones in the half-spaces ($R_1$ and $R_2$).

### 2.2.4    Description of error propagation

As described in section 1.3, the errors in neutron counts follow a Poisson distribution. In this study we exclusively consider large numbers for values of $N$ (i.e. $N \gg 20$, which is consistent with practical CRNS applications). As a consequence of the

240 Central Limit Theorem, we can approximate the errors with a Gaussian distribution and corresponding standard deviations:

$$
P(\lambda) \sim \mathcal{N}(\mu=\lambda, \sigma^2=\lambda^2). \qquad (16)
$$

As the considered errors in the two directional counts ($N_{\mathrm{f1}}$ and $N_{\mathrm{f2}}$) are independent, the superposition of these errors can be described based on Gaussian error propagation:

$$
\sigma_F = \sqrt{\left(\frac{\partial F}{\partial x}\right)^2 \sigma_x^2 + \left(\frac{\partial F}{\partial y}\right)^2 \sigma_y^2}, \qquad (17)
$$

where $F$ denotes a function combining the contributions of $x$ and $y$. In our case, $F$ would be the reconstruction of the count rates from the directional counts (i.e. Eq. (14)).

     When reconstructing the rates in the half-spaces $R_1$ and $R_2$ as described in Eq. (14) and propagating the errors in both directional counts (see Eq. (17)), and performing substitutions (eqs. (4) and (13)), the error in the reconstructed epithermal


counts is:

$$\boldsymbol{\sigma}_{\mathrm{r,total}} = \sqrt{\left(\mathbf{B}^{-1}\right)^{\circ 2} \boldsymbol{\sigma}_{\mathrm{f,total}}^{\circ 2}}^{\circ}$$
$$= \sqrt{\left(\mathbf{B}^{-1}\right)^{\circ 2} \sqrt{\boldsymbol{N}_{\mathrm{f,total}}}^{\circ\circ 2}}^{\circ}$$
$$= \sqrt{\left(\mathbf{B}^{-1}\right)^{\circ 2} \mathbf{B}\boldsymbol{N}_{\mathrm{total}}}^{\circ}. \tag{18}$$

($\circ$ denotes the Hadamard operations, i.e. the square and square root operation are applied element-wise on the vectors and matrices.)

### 2.2.5  *Determining* and *Distinguishing* neutron count rates

As mentioned in section 1.2, directional CRNS is motivated by two aims:

I) *Determining* the neutron count rate for the area the detector is directed at (optionally, also the rate for the complementary area)

II) *Distinguishing* the count rates of the two areas, i.e. detecting a difference in neutron flux.

Both aims can be pursued jointly or separately, however, achieving the one does not necessarily guarantee achieving the other. For example, it might be possible to determine two count rates with high precision, but nevertheless it might be impossible to detect a significant difference between them, when they are (nearly) equal. Conversely, very dissimilar count rates can be discerned, even if their actual value cannot be determined very precisely. Therefore, we look at both of these aims separately. We formalize the determination of count rates (aim I) as being able to confine their 95-%-confidence interval (CI) to a value smaller than a desired precision, expressed as a fraction of the actual value. We chose a value of 5 % for our illustrations, which roughly corresponds to an error of 2 percent-points in volumetric water content for the conversion used by Fersch et al. (2020). For distinguishing two count rates (aim II), we propose they are significantly different, if their difference – regarding the associated uncertainties – is statistically different from 0 with a chosen p-value (0.05 in our case).

### 2.2.6  Selected example values chosen in the analysis

Applying Eq. (18), e.g. using provided R-code, the described analysis can be performed for any combination of parameters of interest, i.e. count rates ($R_1$, $R_2$) or their respective contrast ($\Delta R$), temporal resolution ($\Delta t$), directional contribution ($\eta$) and the fraction of bycatch neutrons ($\epsilon$). However, for illustrative purposes, in the Results section (3.4) we provide some example plots trying to capture the typical ranges of the parameters involved. This selection was guided by a range of diverse settings found in the literature summarized in Tab. 2.

**Fraction of non-epithermal counts ($\epsilon$)**

In the examples, we illustrate the situation for an "bestcase" scenario assuming a low fraction of non-epithermal counts ($\epsilon = 0.1$) and for a "worstcase" setting ($\epsilon = 0.3$).





**Table 2.** Neutron count rates reported in experimental CRNS-studies. Bold entries have been approximated in the by the representative examples.

| source | site | elevation [m] ASL | sensor | count rates $R_{\text{total}}$ [counts h$^{-1}$] | | | max. contrast |
|---|---|---|---|---|---|---|---|
| | | | | mean | min | max | $\Delta R$ |
| Bogena et al. (2013) | humid forest | 595 | CRS1000 | **450** | 410 | 510 | 0.2 |
| Rivera Villarreyes et al. (2011) | agricultural lowland | 84 | CRS1000 | 730 | 521 | 930 | **0.8** |
| Baroni et al. (2018) | agricultural lowland | 60 | CRS1000 | 917 | 726 | 1108 | **0.5** |
| Fersch et al. (2020) | pre-alpine grassland | 595 | CRS1000 | 800 | 550 | 1000 | **0.8** |
| Schrön et al. (2017) | grassland | 78 | CRS1000 | 1500 | 1400 | 1650 | **0.2** |
| Fersch et al. (2020) | pre-alpine grassland | 595 | CRS2000B | **2100** | 1800 | 2500 | 0.4 |
| Schattan et al. (2017) | alpine | 2480 | CRS1000 | 4000 | 2500 | 6000 | **1.4** |
| Fersch et al. (2020) | pre-alpine grassland | 595 | Lab-C | **8000** | 6800 | 9000 | 0.3 |
| Fersch et al. (2020) | pre-alpine grassland | 595 | FZJ-rover | **38919** | 33100 | 44700 | 0.4 |
| hypothetical rover at Schattan's site | alpine | 2480 | FZJ-rover | **144000** | 90000 | 216000 | 1.4 |

**Directional contribution ($\eta$)**

The effect of the directional shielding is expressed by the directional contribution $\eta$, i.e. the ratio between the counts from the area targeted and total counts (see Eq. (1)). In the displayed examples we only consider symmetrical half-planes (see Fig. 3 for the "bestcase" scenario with the perfect shielding ($\eta = 0.72$) and the "worstcase" implementation of the actual shielding ($\eta = 0.61$) as resulted from the neutron simulations (see section 3.3, Tab. 3).

**Fraction of non-albedo neutrons in the directional detector ($\gamma_{\text{s}}$)**

The fraction of non-albedo neutrons, i.e. those without the interaction with the surface, is a function of the hydrogen inventory in the footprint. We estimate respective values based the neutron simulations (see section 3.3). For the "bestcase" scenario, we choose a low value ($\gamma_{\text{s}} = 0.2$) corresponding to very dry conditions; the "worstcase" implementation ($\gamma_{\text{s}} = 0.31$) mimics a very wet environment (see section 3.3 and Tab. 3).

**Overall reduction of count rate due to the directional shielding ($\beta$)**

The directional shielding reduces the total count rate in the directional detector to the fraction $\beta$ (see Eq. (7)). For the "bestcase" scenario, we choose a high value ($\beta = 0.4$) corresponding to very wet conditions; the "worstcase" implementation ($\beta = 0.3$) mimics a very dry environment (see section 3.3, Tab. 4 ).





**Count rate ($R$)**

Count rates depend on site conditions (i.e. incoming neutrons and hydrogen pools) and detector sensitivity. We selected values of 500, 2000, 8000, 40000, and 150000,counts h$^{-1}$ as example values for the count rates registered at the detectors ($R_{\text{total}}$, see (4)).

**Contrast in count rates of the half-planes ($\Delta R$)**

We denote the difference in the count rates in the two half-planes with $\Delta R$, expressed as their difference relative to the lower of the two values:

$$\Delta R = \frac{R_2 - R_1}{R_1} \tag{19}$$

Information on a realistic range of this value would ideally be obtained from spatially distinct sensor locations, e.g. from roving. As this information is unavailable for most considered examples, we use the *temporal* variation of the signal as a proxy, resulting in example values of 0.2, 0.5, 0.8 and 1.4.

**Aggregation time $\Delta t$**

Schrön et al. (2018b) suggests that the typical temporal resolution of standard stationary detectors is in the range of 3 to 12 h, depending on detector technology and site conditions. Bogena et al. (2013) and Fersch et al. (2020) use longer aggregation intervals of 24 h. For our study, we display results for values of 1, 6, 12 and 24 h. Longer aggregation times are not recommended from a hydrological perspective, since they would commonly imply too high a change of the observed variable during that interval (e.g. rainfall or drying and respective change in $R$).

For a subsequent evaluation of the feasibility for a *specific* case, we use the setting listed in line 2 of Tab. 2, i.e. the measurement using the prototype directional detector described in section 1.2, i.e. $R_1$=2100 counts h$^{-1}$, $\Delta R$=0.4.

## 3 Results and Discussion

### 3.1 Neutron simulation

#### 3.1.1 Effect of shielding on count rates

Increasing the thickness of the polyethylene shielding does not only reduce the neutron flux from the non-targeted direction, it also partially reflects neutrons from the other direction and changes the energy response. Furthermore, the moderator itself produces secondary neutrons, which can be regarded as an offset bias. Thus, with the main function of blocking neutrons from the non-targeted directions, a number of secondary effects come along, which slightly change the characteristics of the CRNS probe. Tab. 3 summarizes the effects of adding a directional shielding to a detector. It demonstrates that the fraction of





non-albedo neutrons $\gamma$ is higher for the directional detector than for the unshielded operation. Furthermore, it increases with the amount of hydrogen in the footprint.

Secondly, the total count rate reduction by adding the shield $\beta$ is at least 30 %. For the wetter conditions, it is even higher.

**Table 3.** Effect of directional shielding on neutron counts. Simulated counts for different configurations (volumetric soil moisture ("theta") and shieldings ("no shield "/"directional")). The counts are differentiated in those with / without surface interaction ("albedo"/"non-albedo").

| configuration | | counts | | | fraction of "total" | | | fraction of "no shield" | | |
|---|---|---|---|---|---|---|---|---|---|---|
| $\theta\,[\,\%_{\mathrm{Vol}}]$ | **shielding** | non-albedo | albedo | total | non-albedo ($\gamma$) | albedo | total | non-albedo | albedo | total ($\beta$) |
| 3 | no shield | 5084 | 33959 | 39043 | 0.13 | 0.87 | 1.00 | 1.00 | 1.00 | 1.00 |
| | directional | 2321 | 9564 | 11885 | 0.20 | 0.80 | 1.00 | 0.46 | 0.28 | 0.30 |
| 10 | no shield | 4858 | 24300 | 29158 | 0.17 | 0.83 | 1.00 | 1.00 | 1.00 | 1.00 |
| | directional | 2379 | 8170 | 10549 | 0.23 | 0.77 | 1.00 | 0.49 | 0.34 | 0.36 |
| 50 | no shield | 4820 | 13370 | 18190 | 0.26 | 0.74 | 1.00 | 1.00 | 1.00 | 1.00 |
| | directional | 2289 | 5025 | 7314 | 0.31 | 0.69 | 1.00 | 0.47 | 0.38 | 0.40 |

## 3.2 Angular sensitivity

The CRNS method relies on the principle that the detected neutrons have interacted with the soil of the footprint – usually several times – and thus carry information about its hydrogen inventory. Due to these atmosphere-ground interface crossings the correlation between neutron origin and field of view of the probe is diluted. Most neutron scatterings before detection are located in the direct vicinity of the detector, which to some extent dissociates the vector of detection and the vector to the origin of the neutron. For this reason the shielding does not as effectively filter neutron vectors from remote origins, but its directional specificity is much more pronounced for neutrons with origins in the near-range.

From Fig. 5, which shows the case of a detector in the "bestcase" scenario, we can conclude the following: The largest part of the neutron flux remains undetected (black), due to insufficient energy or being scattered off the detector material itself. Most neutrons counted entered the instrument from a viewing angle corresponding to the open side face (orange). However, their origin only partially lies in that direction. While often being transported 'geometrically' (i.e. directly) to the detector when being released from the soil in the direct vicinity of the instrument (light blue line), more distant origins tend to incur much more directional changes of the neutron. This leads to a flattened angular distribution (dark blue line).

## 3.3 Shielding effect

In the "worstcase" scenario, the insufficient shielding of MeV-neutrons to the sides leads to field-of-view-contamination of approximately 10 % of the signal, mostly due to the limited thickness of the HDPE shielding, see Fig. 6. Compared to the

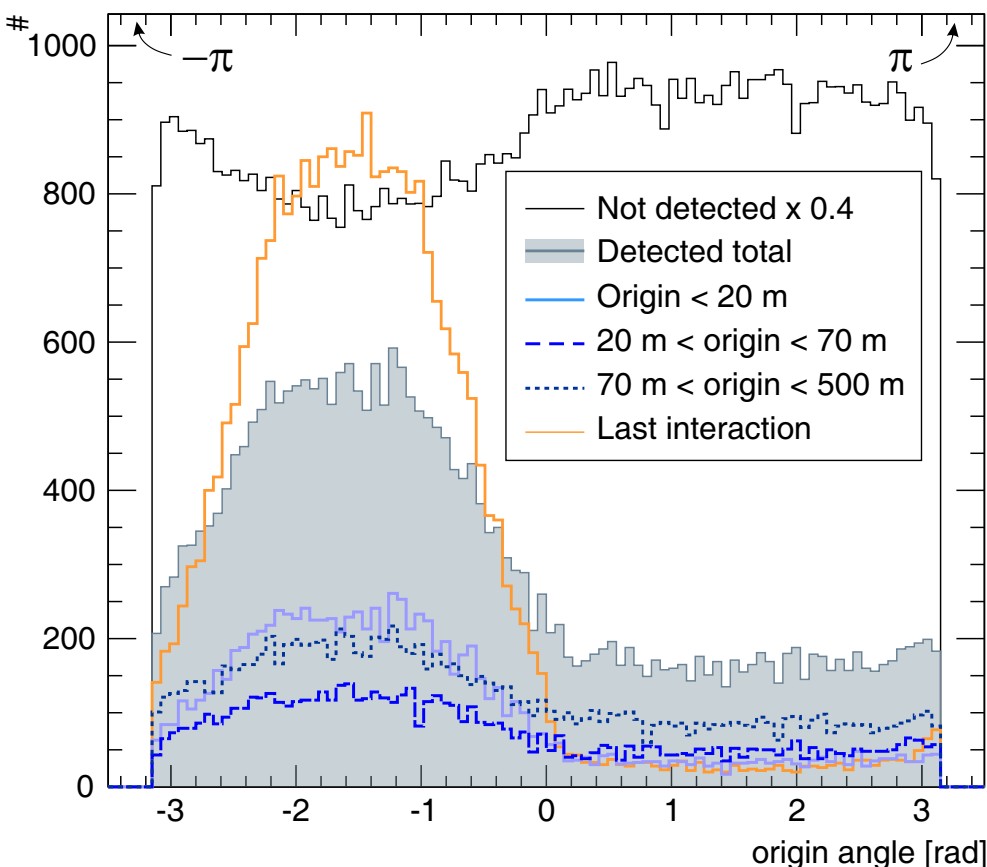

**Figure 5.** Neutron origin angles for the designed directional detector (shielded to all sides except the front which points to $-\pi/2$). The range groups are defined by the distance of neutron origin to the detector, except 'Last interaction', which refers to the last scattering before detection, typically in air. The distribution 'not detected' refers to the total flux through the instrument, please note the different scale.

ideally shielded detector in Fig. 5, the contamination causes an increase of roughly 50 % in the plateau region of undesired angles.

In order to quantitatively assess the directional sensing capabilities we use the directional contribution (see Eq. (1)). We chose as representative target FOVs 90° ($\pi/2$) and 180° ($\pi$). The results are summarized in Tab. 4. For a rather narrow viewing
angle (i.e. 90°) even in an optimistic case less than half of the signal originates from that direction. If the field-of-view limitation is set to a full half-space 60 % of the signal is representative for information from those angles. As the actual detector suffers from a partial leaking-in of MeV neutrons, a near-field blur effect appears: As most neutrons from the direct vicinity are fast the signal contamination due to insufficient shielding is to a large extent carrying information about the local area of the sensor.

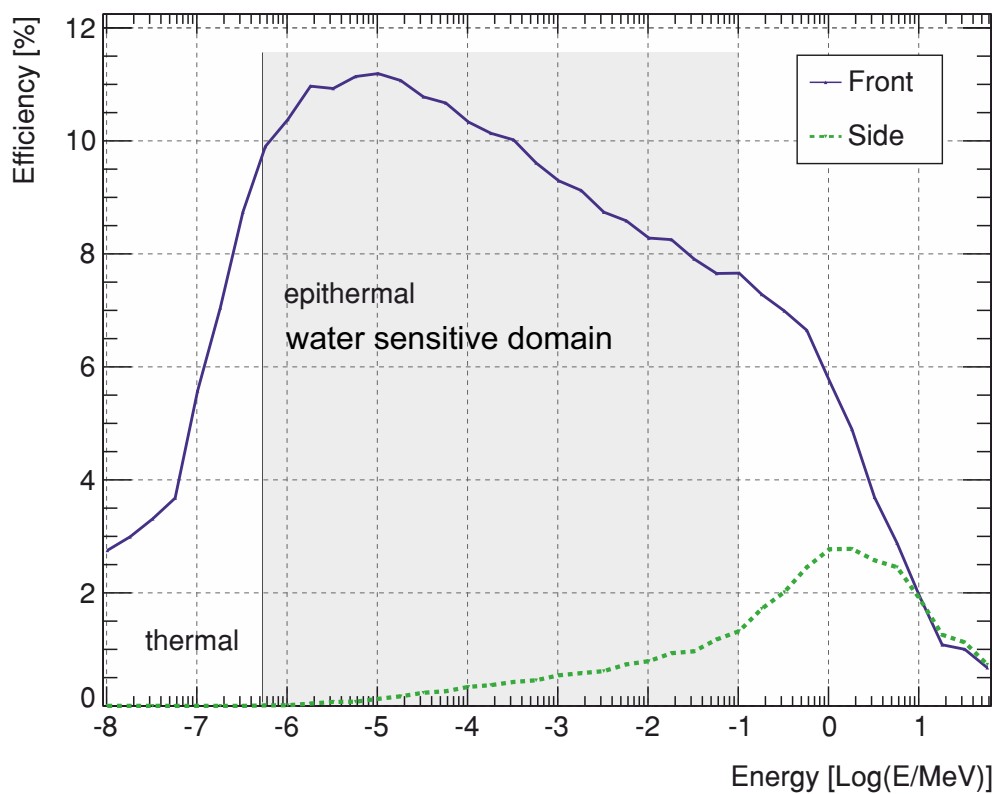

**Figure 6.** Energy response function of the directional detector for the open face ('Front') and a side face. The sensitivity difference between side faces and the one opposed to the open face ("back side") is negligible.

**Table 4.** Directional contribution $\eta$ for the actual detector model and an optimistic case in which the side faces are 100 % impermeable for neutrons. Compare the range groups also to Fig. 5.

| FOV | $\pi/2$ (90°)) | | $\pi$ (180°) | |
|---|---|---|---|---|
| group | act. detector | bestcase detector | act. detector | bestcase detector |
| all | 0.37 | 0.45 | 0.61 | 0.72 |
| < 20 m | 0.40 | 0.52 | 0.63 | 0.81 |
| 20...70 m | 0.36 | 0.41 | 0.60 | 0.68 |
| > 70 m | 0.35 | 0.39 | 0.59 | 0.66 |

### 3.4 Feasibility of directional CRNS-measurements

Based on the presented findings of the neutron simulations, the following analysis has been made to assess the feasibility of the directional detector.



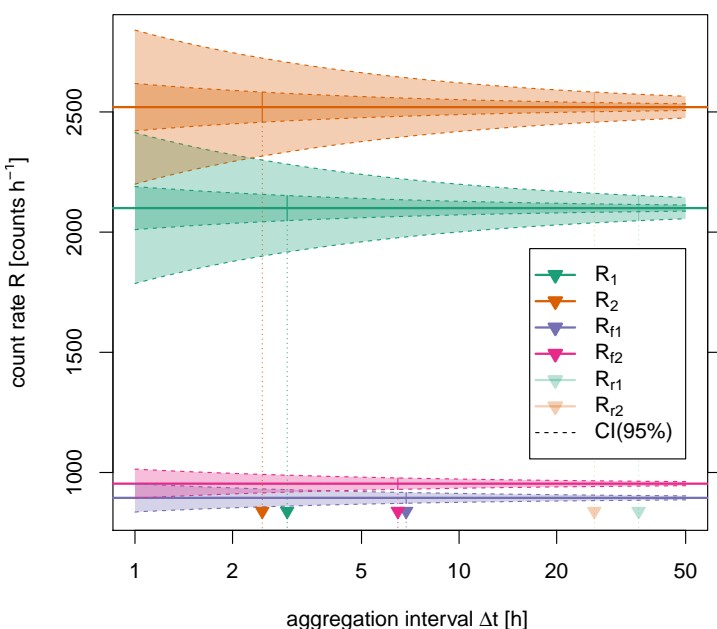

**Figure 7.** Example for 95 %-confidence intervals for the count rates obtained within the half-spaces ($R_{1,\text{total}} = 2100$ counts h$^{-1}$, $R_{2,\text{total}} = 2520$ counts h$^{-1}$, "bestcase scenario", moderate contrast $\Delta R = 0.2$), by the directional CRNS-sensors ($R_{\text{f1,total}}$, $R_{\text{f2,total}}$) and the rates reconstructed from the directional sensors ($R_{\text{1r,total}}$, $R_{\text{2r, total}}$). The vertical arrows indicate the minimum required aggregation time to confine the CI within 5 % of the true value ($\Delta t_{\min}^{\text{determ}}$).

Figure 7 illustrates an example simulation ($R_{1,\text{total}} = 2100$ counts h$^{-1}$, $R_{2,\text{total}} = 2520$ counts h$^{-1}$, $\epsilon = 0.3$). It clearly shows how the rates of the directional sensor ($R_{\text{f1,total}}$, $R_{\text{f2,total}}$) are considerably lower due to the blocking effect of the directional shielding. Concerning the *determination* of rates, the confidence intervals narrow with increasing aggregation time. Notably,
the CI for the reconstructed rates $R_{\text{1r,total}}$ and $R_{\text{2r,total}}$ is always considerably wider than their directly measured counterparts $R_{1,\text{total}}$ and $R_{2,\text{total}}$. So, how much aggregation time is required to determine a count rate precisely? We define $\Delta t_{\min}^{\text{determ}}$ (arrows in Figure 7) as the time when the CI gets smaller than the chosen precision of 5 % of the true value, i.e. 5 % relative precision. We use this time as an indicator for the time beyond which the *determination* of a count rate becomes reasonably accurate. As our focus is on the rates reconstructed from the directional measurements, the following statements refer to $R_{\text{1r, total}}$, the lower
of the two reconstructed count rates, as it has the larger value of $\Delta t_{\min}^{\text{determ}}$ (green arrow in Figure 8 at 36 h).

In the context of *distinguishing* the two count rates, Figure 8 shows that the p-value for comparing two count rates decreases with increasing aggregation time. As one would expect, a statistically significant difference between two rates is more discernible with longer aggregation times. For the directly-measured rates $R_{1,\text{total}}$ and $R_{2,\text{total}}$, this distinction is possible even for



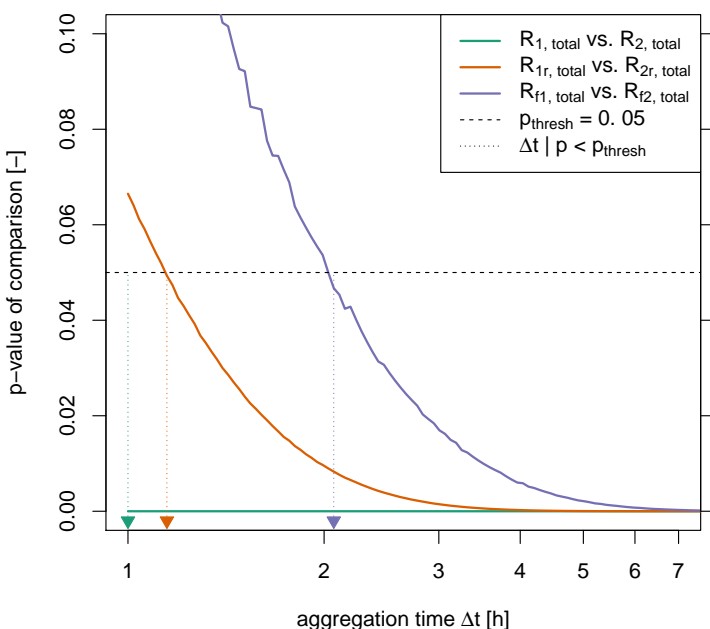

**Figure 8.** p-values for discriminating pairs of measured count rates ($R_{1,\text{total}} = 2100$ counts h$^{-1}$, $R_{2,\text{total}} = 2520$ counts h$^{-1}$, "bestcase scenario", moderate contrast $\Delta R = 0.2$). The vertical arrows indicate the minimum required aggregation time $\Delta t_{\text{min}}^{\text{disting}}$ to statistically *distinguish* two rates at p = 0.05.

the lowest aggregation times, while it takes longer for the reconstructed rates $R_{1r,\text{total}}$ and $R_{2r,\text{total}}$. (Somewhat surprisingly,

distinguishing between $R_{f1,\text{total}}$ and $R_{f2,\text{total}}$ is apparently even harder. The reconstruction of these rates using Eq. (14) evidently increases their signal-to-noise ratio.) How much aggregation time do we need to distinguish two count rates statistically? We choose the time $\Delta t_{\text{min}}^{05}$ as an indicator for the time beyond which the difference between two rates $R_{1r}$ and $R_{2r}$ is significant at the 5 %-level (brown arrow in Figure 8).

     So while an aggregation time of at least $\Delta t_{\text{min}}^{\text{determ}}$ is needed to pinpoint one reconstructed value, we require $\Delta t_{\text{min}}^{05}$ to distin-

guish two rates. These two objectives are different, and we will show that both indicators differ accordingly.

### 3.4.1   A: What temporal resolution can be obtained?

Figure 9 confirms that for higher count rates $R_1$ and $R_2$, less time is required to obtain a robust value from the directional measurements, i.e. confining the CI to less than 5 % of the actual value. The respective contrast between $R_1$ and $R_2$ is of relatively small effect for higher count rates, but makes a difference for lower ones. For these, somewhat counter-intuitively, a

higher contrast requires longer aggregation times. This may be explained by the fact that these higher count rates are associated



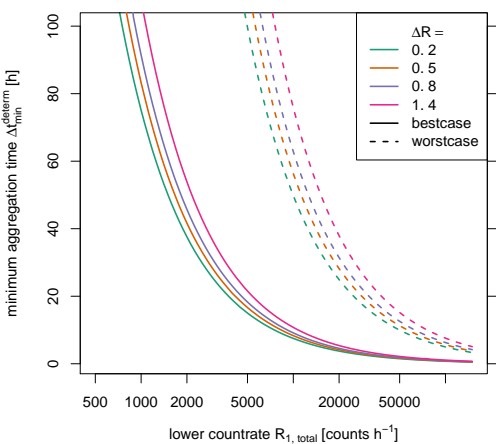

**Figure 9.** Minimum aggregation time $\Delta t_{\min}^{\text{determ}}$ required to obtain the reconstructed count rate $R_{1r}$ from the two directional measurements with a CI smaller than 5 % of the actual value.

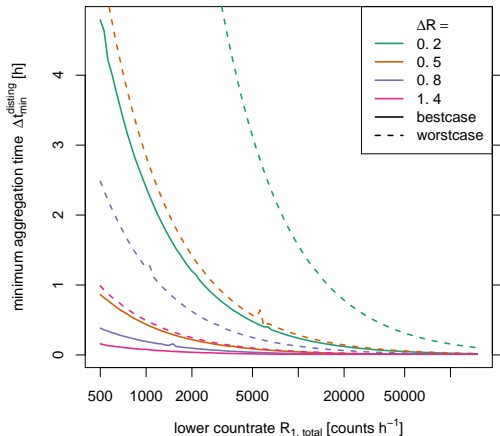

**Figure 10.** Minimum aggregation time $\Delta t_{\min}^{\text{disting}}$ required to statistically separate the reconstructed count rates $R_{1r}$ and $R_{2r}$ from the two directional measurements with p-value $< 0.05$.

with larger absolute errors. When mixed with the weaker signal (see Eq. (14)), they deteriorate its robust reconstruction. Consequently, a higher contrast in the rates aggravates the reconstruction of the lower one.

Conversely, Figure 10 demonstrates that the statistical difference between $R_{1r}$ and $R_{2r}$ is easier to detect with a higher contrast between the two rates. This phenomenon effectively equates to the dilemma that the contrast in the count rates has the





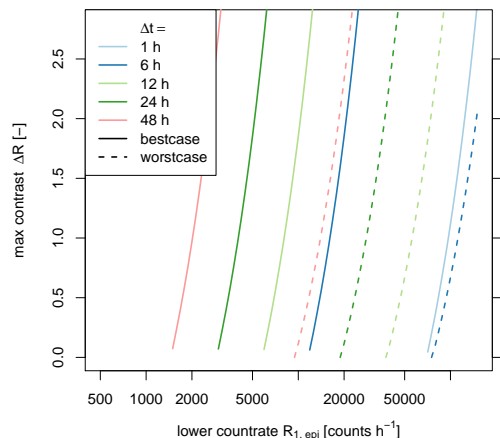

**Figure 11.** Maximum possible contrast $\Delta_R$ in reconstructing the count rates $R_{1r}$ from the two directional measurements with a CI smaller than 5 % of the true value.

opposite effect, depending on whether we look at the precise *determination* of $R_{1r}$ (benefits from low contrasts) or the statistical *distinction* between the two rates $R_{1r}$ and $R_{2r}$ (benefits from high contrasts). This distinction is apparently much more feasible within reasonable aggregation times: in the "bestcase" scenario, hourly resolution can be achieved from count rates of above approx. 3000 counts h$^{-1}$ even for low contrasts. The "worstcase" scenario increases the aggregation times by about factor 7.

For our example case, we can conclude that the precise *determination* of the count rates is hardly feasible even for the "bestcase" scenario. Aggregation times exceeding at least 36 h are beyond the typical requirements in applications. Aggregation times of 24 h and less can only be achieved with count rates of more than 3100 / 20700 counts h$^{-1}$ for the "bestcase"/"worstcase" scenario. *Distinguishing* the rates in the two half planes, however, could be possible for aggregation times in the order of hours with even higher potential for stronger contrasts.

### 3.4.2   B: What spatial contrast in the count rates can be resolved?

Figure 11 again illustrates the phenomenon mentioned in the previous section: With higher contrast in the signal, estimating the weaker signal with the required precision gets more difficult (i.e. requires longer aggregation times or higher count rates). For reproducing values with high contrast ($\Delta R = 1.4$) in daily resolution, count rates $R_1 > 4000$ counts h$^{-1}$ are required for the "bestcase" scenario. For the "worstcase" scenario, $R_1$ must be larger than 40000 counts h$^{-1}$ for this purpose.

Conversely, higher contrasts allow the statistical distinction of the two reconstructed rates also for lower count rates and aggregation times (see Fig. 12). According to the "worstcase" scenario, even relatively low contrasts ($\Delta R = 0.2$) can be detected with the lowest considered count rates, if aggregation times are slightly higher than 24 h.





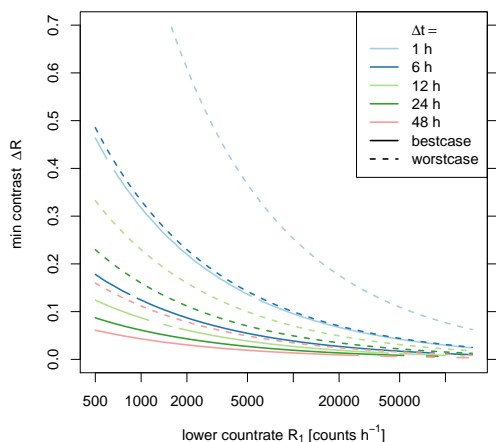

**Figure 12.** Minimum contrast $\Delta_R$ required to statistically separate the reconstructed count rates $R_{1r}$ and $R_{2r}$ from the two directional measurements with p-value < 0.05.

For our example case, we have already shown that the *determination* of the count rates is hardly feasible even for the "bestcase" scenario. However, with aggregation times of 24 h, *distinguishing* rates with contrasts lower as 0.2 could be possible even for the "worstcase" scenario.

### 3.4.3 C: What count rates are required to yield robust estimates?

Figure 13 again stresses the need of high count rates to reconstruct the target count rates with the chosen precision. Specifically, to compute those rates in the "bestcase" scenario at the daily resolution, the minimum count rate must be well above 3000 counts h[-1] for contrasts lower than 0.2. For the "worstcase" scenario, this value increases to over 20800 counts h[-1]. As noted before, higher contrasts require even higher count rates (i.e., > 4500 / 31000 counts h[-1] for the cases above) or longer aggregation times.

Concerning the statistical discernability of $R_{1r}$ and $R_{2r}$, Figure 14 suggests that already low count rates allow the two reconstructed count rates to be distinguished. For the "worstcase" scenario at the aggregation interval of 1 h, count rates of at least 15600 counts h[-1] already allow resolving contrasts as low as 0.2. For the higher contrasts ($\Delta R = 1.4$), count rates of roughly 490 counts h[-1] would suffice.

Thus, for our example case, we would require at least 3100 counts h[-1] to successfully reconstruct both rates at 24-h-resolution. The current count rate of 2100 counts h[-1] calls for aggregation times of at least 36 h. Merely distinguishing the rates $R_{1r}$ and $R_{2r}$ is possible with the given count rate at 1-h-aggregation time.





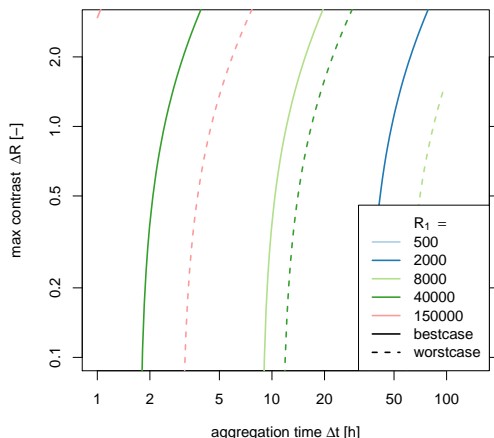

**Figure 13.** Maximum possible contrast $\Delta_R$ to allow the reconstruction of the count rate $R_{1r}$ from the two directional measurements with a CI smaller than 5 % of the true value with the aggregation time $\Delta t$.

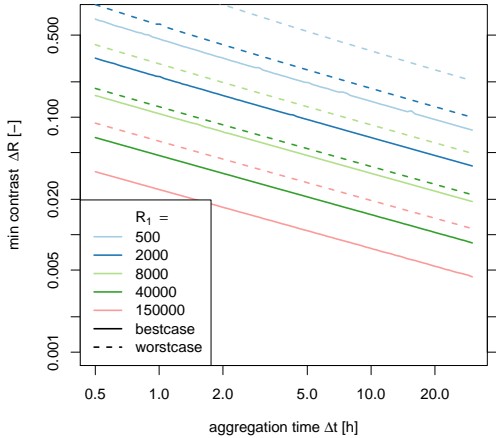

**Figure 14.** Minimum contrast $\Delta R$ required to statistically separate the count rates $R_{1r}$ and $R_{2r}$ reconstructed from the two directional measurements with a p-value < 0.05.





## 3.5  Limitations and Outlook

- This analysis bases on the geometries of a specific directional detector. This prototype was designed with pragmatic considerations. Other designs (e.g. larger planar shieldings) may provide superior characteristics, namely $\eta$, which could be assessed by further neutron simulations.

- Fig. 3 suggests a setup of two independent detectors facing opposite directions. In this case, an imperfect calibration of their sensitivity can constitute another substantial device-specific error source, further aggravating the separation of the signals. Alternatively, a single sensor with temporally varying orientation (i.e. actual "scanning" as presented in section 1.2) could be used. While this eliminates the issue of imperfect calibration, it implies that all the computed aggregation times must be doubled, as the single sensor can cover each direction only half of the time.

- The presented computations assume constant neutron flux rates. Evidently, this assumption is less realistic with increasing aggregation times: On the one hand, incoming neutron flux is subject to variation, though usually moderate, on the other hand, changes in the hydrogen pool within the footprint, namely due to hydrological processes, will add more variability to the signal, effectively increasing its error and aggravating the determination and discrimination of count rates.

- The directional contributions obtained from the neutron simulations apply to a setting with selected values for fixed hydrogen inventories. Increasing this inventory, would decrease the sensor footprint. As the angular specificity decreases with distance to the sensor, we might expect somewhat improved directional contributions. However, in practice, an increased hydrogen inventory (i.e. from soil moisture and biomass) will usually also incur higher air humidity (deteriorating angular specificity) and reduced count rates, which counteract this effect. This is similar to the adverse effect of road-construction material on roving CRNS measurements (Schrön et al., 2018a). In-depth neutron simulations need to be used to clarify this issue.

- Likewise, our setup also assumed spatially homogeneous hydrogen pools. Deviations from this assumptions, especially close to the sensor, will have a pronounced effect on the recorded signal due to the high sensitivity of the sensor in the short range. These effects will be detrimental to the reconstruction of representative count rates for the half-planes, unless the interpretation is restricted to this very proximity of the sensor.

- With disparate flux rates $R_1$ and $R_2$ two additional concurrent effects will occur: If $R_1$ increases (e.g. less moisture in $A_1$), we would also expect more $A_2$ neutrons to be scattered back to the detector from $A_1$, because thermalization is reduced there. This deteriorates the directional contribution $\eta$ for the reconstruction of $R_{r1}$ as we are getting more neutrons from the "wrong" direction. Conversely, the increased $R_1$ will tend to be more directional when coming from the area of less hydrogen (i.e. less scatter), in turn increasing the directional contribution for $R_{r1}$. Further neutron simulations are required to clarify which of these effects dominate and if they would notably influence $\eta$.





– We presented examples for a field-of-view 180 degrees. While smaller angles would be desirable for higher spatial resolution, such signals from smaller angles will be more difficult to resolve, due to the reduction of the count rates (smaller $\beta$). Although the abovementioned methodology remains the same, the matrices $\mathbf{A}$ and $\mathbf{B}$ are no longer symmetric.

– In the choice of example values, we also included some count rates obtained with setups of roving CRNS. These setups consist of a larger number of counting tubes. Consequently, they are considerably bulkier and could not be equipped with a shielding in the described dimensions, unless advances in sensor technology provide significantly smaller detectors. Even if such a shielding could be scaled up, the resulting weight would pose a severe challenge for realizing a practicable *rotating* sensor platform, thus calling for two complementary, non-rotating sensors instead.

– This study exclusively considered count rates as the target observation variable. However, for application, count rates are merely a proxy which need to be converted to the actual quantity of interest, namely soil moisture, snow or biomass. The relationships used in such conversion (e.g. Desilets et al. (2010)) are not linear; instead, they saturate with increasing hydrogen pools and low count rates. This translates to an increased sensitivity of the target variable (e.g. soil moisture) for these low rates. This behaviour, therefore, amplifies the characteristics demonstrated in this study: considerably poorer applicability of directional CRNS-measurements for lower count rates. Combining the presented approach with the one of Jakobi et al. (2020) would allow the direct quantification of uncertainty for the target variable.

– The directional contribution $\eta$ depends on the transport characteristics of the neutrons from the measured object to the sensor, and the properties of the shielding. The former is governed by the surrounding medium (i.e. air pressure and humidity) and is beyond control in monitoring situations. The shielding, however, can be modified without theoretical limitations. Although obtaining a narrow FOV is still unfeasible because of the above-mentioned transport characteristics, hemispherical blocking could be increased to a large extent. It is constrained, however, by practical issues such as size, weight and price. Directional shielding larger than a few metres would hardly be practicable (transport, visual impact, wind stress); extending the shielding below the soil surface is hardly feasible, a freely rotating setup poses even stronger limits. On the other hand, a concomitant use of two sensors could potentially use the same shielding. An enlarged version of the shielding would presumably also have higher directional contribution. The presented methodology could help to find reasonable compromises.

– In our calculations, for the "bestcase"/"worstcase" scenario we use fixed values for the fraction of "bycatch" count ($\epsilon$) at $10\,\%$/$30\,\%$. This assumption implies that these adjacent energy levels (thermal and fast neutrons) show a similar sensitivity to hydrogen. In reality, this sensitivity is considerably smaller and different, but has been exploited in some studies (e.g. Baatz et al. (2015); Tian et al. (2016)), effectively reducing uncertainties by increasing the rate of usable counts $R_{\mathrm{epi}}$. However, measuring with only single moderated detectors does not allow for such a separation of the signal in terms of energy levels. Instead, the actual value of $\epsilon$, being a function of hydrogen pools and chosen energy cutoff thresholds, poses another considerable source of uncertainty, which we neglect completely in this study by assuming a



fixed value. A similar effect can be expected for the fraction of non-albedo neutrons $\gamma$: Its range was considered in the scenarios, however, its actual functional dependency on the hydrogen inventory ignored in the analysis.

– As more angle-specific option for the shielding, micro-channel plates have shown favourable directional characteristics (Tremsin et al.). However, their limitation to thermal neutrons, considerable costs and the remaining problem of
non-geometric transport makes them unfeasible for CRNS.

– Materials with low hydrogen content and a high scattering/absorption ratio with respect to the nuclear cross sections, for example graphite or quartz, can be used as reflectors towards the inside for directions, which are not of interest. Such reflectors can be used either on the insides or outsides of the moderator or in 'shark' geometry between HDPE plates.

## 4   Conclusions

This study combined neutron simulations with an analytical assessment of the directional specificity of epithermal neutrons, and the potential for directional measurements in environmental monitoring.

The neutron simulations revealed the relatively low correspondence of incidence angle and angle to origin of the neutrons arriving at a CRNS-detector. This is a direct effect of the non-geometrical (i.e. not direct) path of the neutrons, being subject to multiple collisions. Consequently, the correlation of these angles gets lower with increasing distance of the origin (i.e. the
location of its conversion to epithermal). Consequently, blocking certain incidence angles of a sensor only yields a limited angular specificity: For the investigated geometries of a directional shielding as a half-open rectangular box, the directional contribution was in in the range of 60 to 80 % for a target FOV of 180°. Moreover, this additional angular shielding reduced the overall count rates to about 30-40 %.

Based on these findings, the subsequent analytical analysis focused on the feasibility of *determining* and *statistically distin-*
*guishing* the count rates from two adjacent half-spaces by reconstruction from measurements of directional sensors. While the former benefits from a low contrast in count rates, the latter is aggravated by it. Both aspects profit from high count rates and longer aggregation intervals.

With the analyzed setup and reasonable count rates, the accurate reconstruction of the two count rates is hardly feasible with less than 24 h of aggregation time, given detectors with conventional sensitivity. Thus, it seems of little value in environments
where variability needs to be resolved at this time scale. While a substantial increase in detector sensitivity might address this issue, such an increase typically comes with higher costs and much larger detector sizes and hence an unfavorable increase in the dimensions of the required shielding.

The mere distinction of two rates, however, is more feasible and, even for moderate count rates and contrasts, perceivable at a resolution of a few hours. The effort of directional measurements for the mere purpose of distinction might appear somewhat
incommensurate. Yet, the gain in information might be very relevant from a hydrological point of view, e.g. at the borders between grassland and forest which might experience a reversal of horizontal sol moisture gradients in periods of drying.



Progress in detector technology and optimizing the shielding towards wider FOVs but more specificity could alleviate some of the restrictions and make directional scanning a useful tool for tailored use of CRNS.

*Code and data availability.*

The source code for the neutron simulation model URANOS is freely available from https://www.ufz.de/uranos.

The plots of section 3.4 were produced with R (R Core Team, 2020). The source code is provided on Github (Francke, 2021), from where it can also be run directly via the a web browser (https://github.com/TillF/directional_CRNS).

*Author contributions.*

TF and MH designed and conducted the analytical assessment and drafted the manuscript. MK conducted the neutron sim-
510 ulations. SO initiated studying the potential of directional CRNS measurements. CB and SO designed and built the prototype sensor. MS provided crucial input for the study design. All authors contributed to writing and revising the manuscript.

*Competing interests.*

Markus Köhli holds the position of CEO at StyX Neutronica GmbH.

*Acknowledgements.* This research was funded by the Deutsche Forschungsgemeinschaft (DFG, German Research Foundation) – project
357874777 of the research unit FOR 2694 "Cosmic Sense".



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
