# Peer review of "Assessing the feasibility of a directional CRNS-sensor for estimating soil moisture"

_Geoscientific Instrumentation, Methods and Data Systems, 2021_

## Referee Comment (RC3)

Review of MS No.: gi-2021-18
Francke et al. Assessing the feasibility of a directional CRNS-sensor for estimating soil moisture

General comments

The Authors present an interesting and valuable study on the feasibility of directional cosmic-ray neutron sensing (dCRNS). This scanning method could help the interpretation of the signal at specific sites with e.g., land use patches and it could provide some new opportunities for integrating the method for new applications. The manuscript is generally well written and clear and the analysis is based on several numerical simulations. For these reasons I think the study can be a valuable scientific contribution. I have however three main concerns that I think should be considered: [1] to add the experimental results; [2] to improve the quality of story-telling and [3] to strengthen the scientific value of the study with a more clear outlook. Despite these might require some additional work, I believe that the Authors will be able to well address my concerns. Some few specific comments are listed below that I hope can also help the overall improvements of the manuscript.

General comments

[1] At the beginning of the manuscript a direction CRNS sensor has been introduced (fig1). Simulations are based on this sensor design. Surprisingly for me, no experimental results has been shown and discussed in the present study. Why that? I believe that some data have been collected since it is stated that this prototype has been constructed since 2018 (L69). I think adding experimental results would strengthen the value of this study.

[2] The manuscript well covers and discusses the methodology, the results and the limitations of the study. Despite I do not see any critical missing parts, I found the story line not always clear or a bit difficult to read. I think the manuscript could be improved by re-organizing especially the method section in a more synthetic structure that can be appreciated and understood even without reading all the details. A table summarizing the different scenarios simulated could also help the readings.

[3] The final result seems to be that scanning CRNS is hardly feasible and only strong soil moisture changes and patches could be detected with long integration time. Despite I believe that scientific papers should also present and discuss "what does not work" to provide guidelines for further studies, I usually expect that from lab-field studies where it is hardly possible to repeat or to increase the numbers of experiments. In contrast, all the results of the present study are based on simulations. The Authors well discuss some limitations on the numerical settings and they often acknowledge that further simulations could shed lights on how to design an effecting settings (e.g., L411, L428, L438). The Authors conclude for instance that progress in detector technology and optimizing the shielding towards wider FOVs but more specificity could alleviate some of the restrictions and make directional scanning a useful tool for tailored use of CRNS (L501-502). I believe that the scientific value of the present study should be improved not speculating on that. Instead few specific additional simulations could support the design of a proper scanning CRNS in the future. Specifically, I personally suggest to consider a hypothetical very high sensitive sensor with a cone-shaped shielding (smaller angles) that has been shown in literature to be able to discriminate directional neutron fluxes but not in the context of soil moisture observations (see Becchetti et al, 2015). Alternative (or additional) settings based on the experiences of the Authors could be also considered. Despite I agree (see your comments at L442) that these set-ups could be not very practical nowadays considering among others the need of high sensitive (and probably expensive) sensors, it could be probably feasible in the near future considering

current sensors developments. At such, the present study would provide a strong bases for further sensor development. At the present status, this is not the case.

Specific comments

L66. As far as I have understood, side-shielding has also been suggested for addressing neutrons form below-sensor in the so-called "road effect" by Schrön et al. 2018? If this is the case it is worth extending here the discussion with this example and clarifying that directional shielding has been presented for discriminating neutron fluxes in the vertical directions but so far not focusing on lateral side-directions.

Schrön, M., R. Rosolem, M. Köhli, L. Piussi, I. Schröter, J. Iwema, S. Kögler, et al. "Cosmic-Ray Neutron Rover Surveys of Field Soil Moisture and the Influence of Roads." *Water Resources Research*, May 14, 2018. https://doi.org/10.1029/2017WR021719.

L69-80. These descriptions could better fit the methodology section.

Section 1.3 fits better the methodology section, in my opinion.

L117-121. Based on this explanation, I would have expected that directional shielding is not feasible.

L179. Remove "it"

L182. There is a quite long discussion on non-epithermal neutrons. Most of this contribution, as far as I have understood, comes from thermal energy neutrons (20%). But it is then stated the Thermal neutron transport was disabled for reasons of computational speed. I'm a bit confused by the importance of this discussion and the detailed description at L101-113.

L186. Remove one "the"

L310. As far as I have understood, currently all are neutron simulations. Why then the need of the title section 3.1?

L487. I suggest specifying full name in the conclusions

L501. I strongly encourage the Authors to provide more insights on how optimizing the shielding (see general comment [3]).

---

## Author Comment (AC1)

**Interactive Discussion: Author Response to Referee #1**

**Assessing the feasibility of a directional CRNS-sensor for estimating soil moisture**

Till Francke et al.

*Geoscientific Instrumentation Methods and Data Systems, Discuss.*, `doi:10.5194/gi-2021-18`
* * *
**RC:** *Reviewer Comment*,     AR: *Author Response*,     ☐ Manuscript text

Dear Referee,

thank you very much for your positive comments and constructive suggestions to our manuscript. We very much appreciate the time and effort that you have invested in your report.

Based on your report, we have already started to revise our manuscript. Please find below our detailed responses to all the points you have raised in your report. We will continue to address these points, and are confident that the manuscript will substantially improve as a consequence. Yet, the final implementation of changes will also depend on another referee report that is still to be submitted in the interactive discussion.

Kind regards,
Till Francke
(on behalf of the author team)

**1.1. General comments**

**RC:** *Review of manuscript titled "Assessing the feasibility of a directional CRNS-sensor for estimating soil moisture". The idea is fantastic and very much needed. I think this is study that will gain the interest of the research community. The main limitations of the manuscript are:*

- the assumption that readers are familiar with the cosmic-ray neutron sensing technology for measurement of soil moisture.

**AR:** We indeed consider our manuscript to address readers familiar with the basic concept of CRNS. However, we will add some introductory information in this regard to section 1.1.

**RC:** *- The manuscript is dense and the logical arrangement of ideas together with some limitations in the English language makes this paper difficult to read. It starts by describing a previous directional sensor, but then it shifts the attention to simulation work, without interconnecting both. It seems to me that the manuscript could be much shorter and more straight to the point*

**AR:** We are confident that the reviewers' suggestions (below) will help to improve the legibility of the manuscript.

The description of the existing directional sensor is meant as a justification why we chose to simulate this very realization of a shielded instrument as an example case, as other designs are likely imaginable. We will strengthen this reasoning and interconnecting both parts.

**RC:** *At time it seems that this manuscript could have been partitioned in two different manuscripts parts: a more theoretical description of the error sources, propagation, and computation of directional neutron counts, and a second part about testing the quantitative reasoning with a modeling effort.*

AR: These two parts are actually already distinguished in the "Method" subsection (2.1 and 2.2.) and the corresponding "Results and Discussion" subsections (3.1. and 3.2): *.1 deals with the numerical modelling, necessary to understand the relevant effects and determine actual parameter values for describing them (e.g. $\beta$), while *.2 generalizes these findings to a broader range of settings in terms of count rates, signal contrast and aggregation times. We will improve the description of these two pillars in the introductory sentences of the "Methods" section, and try to clarify the fundamental link between the two.

**RC:** *The "Limitations and Outlook" section contains valuable information for the research community. I suggest condensing this section into a fewer, but still relevant, number of points, or merging the points into a narrative. With so many bullet points I question the usefulness of the current layout. Even this section could be interpreted as a potential technical note or manuscript presenting and discussing limitations and opportunities of directional detectors.*

AR: We agree that the current layout is unfortunate. We will restructure the section by grouping specific items in meaningful sub-topics, e.g. "Simplifications", "Parameters used in the study" and "Potential for further instrumental improvements".

**1.2. Specific comments, details**

**RC:** *Line 14. Define again the abbreviation for Cosmic Ray Neutron Sensing (CRNS) in the manuscript narrative.*

**RC:** *Line 14-15: The first two sentences can probably be merged into one. Consider the following alternative: "Cosmic Ray Neutron Sensing (CRNS) has been widely adopted in the past decade to measure soil water content in environmental sciences" or this one "In the past decade, the adoption of Cosmic Ray Neutron Sensing (CRNS) has increased considerably to measure soil water content in hydrological, agricultural, and environmental research applications (Zreda et al., 2008)"*

**RC:** *Line 15. Unclear statement "in both research and application". Research could be basic or applied, so its unclear whether the authors refer to applied research or applications beyond research (end user or consumer applications). Please clarify.*

**RC:** *Line 17. Remove the first "or" so that "..., to support irrigation management..."*

**RC:** *Line 22: Re-word to "a depth of a few decimeters" It may be good to provide a quantitative range. It seems that the typical sensing depth oscillates between 10 and 40 cm depending on the soil moisture conditions.*

AR: –> All suggestions above have been / will be implemented.

**RC:** *Line 25. This paragraph probably needs more context and depth. At this point it's not obvious that the CRNS is omnidirectional, so it may be good to state that this is the nature of current measurements and available instruments. I suggest briefly expanding more on the issue of instrument omnidirectionality (why is it this way?) and other factors such as soil spatial heterogeneity that can create soil moisture spatial patterns. I think this paragraph would be essential to understand the motivation of the study.*

AR: We will extend the paragraph with these suggestions.

**RC:** *The "larger spatial support" compared to what measurement? point-level?*

AR:    The comparison refers to classic point scale measurements. The sentence will be modified accordingly.

RC:    *Line 30. Specify that the "energy level separation" is for the spectrum of cosmic-ray neutrons.*

RC:    *Line 33. Consider a better transition here. For instance: "An approach to reconstruct sub-footprint patterns in soil moisture consists of using a dense.... (Heistermann et al., 2021). Although..."*

RC:    *Line 39. Consider merging this paragraph with the previous since it's a continuation of the same argument.*

RC:    *Line 44. I would like to suggest the use of the following terms: "Directional neutron sensing" or "Cosmic-ray Directional Neutron Sensing"*

RC:    *Line 48. There seems to be a typo here: "and/and"*

RC:    *Line 50. Consider replacing "defense" with military applications*

RC:    *Line 51. There is an extra ")"*

AR:    –> All suggestions above have been / will be implemented.

RC:    *Line 55. This paragraph seems too short at only two sentences long. I suggest detailing the type or energy level of the incoming neutrons. It's unclear from the text whether the authors are referring to thermalized or epithermal neutrons. It will also be important to highlight that the term "directional" in the context of this paper mostly applies to the horizontal plane.*

AR:    The paragraph has been merged with its precursor. However, we do not think that in this context the energy level of the neutrons is of relevance and prefer to keep the message of the paragraph simple. A note on focusing on the *horizontal* aspect has been added to the end of section 1.2.

RC:    *Line 59. Spell out "PE" (polyethylene?). Was this a high-density PE?*

AR:    PE spelled out. Mitrofanov et al. just report about the use of "polyethylene", leaving it open whether it was HDPE or not.

RC:    *Line 61. How is Ntotal observed or computed? Is this the total considering the shielding or the total without it? I assume that for simultaneous collimation efficiency you need two devices or alternatively the same device with and without the shielding over short periods of time.*

AR:    The description was ambiguous and modified to

> [$\eta$] being the ratio between counts from the targeted FOV $N_{\text{FOV}}$ and the total counts $N_{\text{total}}$ registered by the detector, i.e. counts from any direction.

Thus, $N_{\text{total}}$ is the observable variable, which can directly be measured by the directional detector. No concomitant operation of an unshielded detector is required.

RC:    *Line 70. Please, clarify whether the unshielded side means less shielding to avoid confusion with a bare detector. For instance, the CRS 2000B already includes some shielding to attenuate neutrons. This means that the authors added an extra layer of shielding.*

AR:    We added the sentence

> This shielding is independent of the moderator already in place used to thermalize the neutrons.

**RC:** *Line 73. Replace 2Pi by 360 degress. What do the authors mean with "flexible"? What is the integration time at each angle? or is it a continuous scanning? I think that a few more details will allow readers to reconstruct the idea and reproduce the study.*

AR: Changes made and requested information added:

> This could be operated to cover the full 360° periphery ($2\pi$) in flexible angular sections with configurable integration times at selected positions, and thus allow for measurements with variable directions producing time-series of count rates for different FOV in the footprint.

**RC:** *Line 99. Would this bycatch flux be similar regardless of the gas used in the detector ($3$He vs $^{10}$BF3)?*

AR: The mentioned 'bycatch' depends to a minor extent on the gas type, gas pressure and moderator configuration. The deviations are insignificant for the purpose of the discussion.

**RC:** *Line 109. Emphasize that these settings are in the detector electronics. Can you add the model of the neutron pulse analyzer from Quaesta?*

AR: We changed the wording to "... threshold settings in the electronics " to underline the fact. There are various forms of non-neutron signals. Some of them can be cut by setting lower and upper thresholds on the detected pulse energy. In the case of radioimpurities their signatures are often indistinguishable from neutrons in case their energy lies in the range of those of neutron conversion products. With usual electronics, independent of the manufacturer, such cannot be discriminated against. They form a plateau 'under' the neutron events.

**RC:** *Lines 126. The wording in this sentence implies that there is an "unshielded side", which most people will take as a bare detector, which I suspect is not the case here.*

AR: To avoid this potential confusion we have added the following sentence to the end of section 1.1.:

> The term "shielding" is often used to denote parts around the detector to moderate ("thermalize") high energy neutrons to be detectable. Here, we will use the term "moderator" for this component, while components aiming to achieve directionality are referred to as "shielding".

**RC:** *Figure 2. I like this figure. To make it a bit more explicit it will be good to add in the description the meaning of the shaded area. Are the arrows pointing in the "increasing" direction? Perhaps adding a "+" and "-" symbols could help the reader. Will it be possible to add magnitudes to the axes and denote with a point the selected combination for this study?*

AR: The shaded areas have no actual meaning apart from being not part of the inner triangle. To avoid this confusion, we have modified the figure, its caption and added "+" and "-".

[Figure]

Figure 1: "Tradeoff-triangle" in directional CRNS-measurements as ternary plot: for a given combination of two of these parameters, the third parameter must be adjusted accordingly to obtain the requested accuracy. The inner triangle (grey) illustrates the parameter space for an increase in required accuracy. The straight outline here is only chosen for the sake of simplicity and could be curved instead.

We intend this figure merely as a conceptual illustration of the trade-off problem. Thus, we'd rather refrain from adding actual numbers. Moreover, the performed multi-dimensional analysis of each of the three factors would result in multiple inner triangles, which would unnecessarily complicate the figure.

**RC:** *Line 129. Unclear whether the authors are referring to the area in the FOV (i.e. area of the open side of the instrument) or the horizontal area part of the sensing footprint.*

AR: We have added a reference to Figure 3 to make clear that area refers to the part of the footprint.

**RC:** *Line 141. While in situ experiments controlling the neutron flux are impractical, in situ validation of the approach is not. For instance, a field validation could be done by intensively measuring soil moisture. For this to work a field with a known directional variability may need to be selected. I just wanted to emphasize that in situ validation may not impractical (in terms of measuring soil moisture).*

AR: We agree that controlling the flux rates in two half-spaces via adjusting the soil moisture is, in theory, conceivable. However, judging from our experience with much simpler efforts to reliably ensure (let alone adjust and hold) the soil moisture to desired values in test sites of adequate size, we replaced the term "impracticable" with "difficult to implement".

**RC:** *Line 149. I like the first question, but it may need to be formulated differently. As it stands now it will be more relative to the ability to regulate the rotary mechanism than the detector itself.*

AR: We posed the question in the most general way. As the results later will show, the required aggregation times

are in the range of hours, so the velocity of the rotary mechanism is by no means a limitation. However, to avoid the confusion, we have replaced the term "scanning" (implying continuous rotation) by "directional measurement" (measurement with variable, but temporally constant orientation) throughout the manuscript.

RC:   *Line 150. The authors vaguely described what is meant with "contrast". Perhaps a bit more background on this metric would be helpful. Most people are familiar with count rates and integration time, perhaps less familiar with signal contrast (unless the authors are referring to signal-to-noise ratio)*

AR:   The description of "contrast" three paragraphs above has been extended.

RC:   *Line 151. What is a robust estimate? It would be nice to provide a more objective metric, perhaps in terms of error in volumetric water content (or the equivalent neutron count).*

AR:   We deliberately used the general expression "robust metric" here, as robustness can be related either to *distinguish* or to *determine* rates, as explained in the sentence following question C.

We also strictly limit the analysis to the count rates, not soil moisture, for the reasons stated in the "Limitations" section:

> This study exclusively considered count rates as the target observation variable. However, for application, count rates are merely a proxy which need to be converted to the actual quantity of interest, namely soil moisture, snow or biomass. The relationships used in such conversion (e.g. [**?**]) are not linear; instead, they saturate with increasing hydrogen pools and low count rates. [...] Combining the presented approach with the one of [**?**] would allow the direct quantification of uncertainty for the target variable.

RC:   *Line 165. Is the stretch of 600 m in all directions? or just in the horizontal direction?*

AR:   600 m is meant as the minimum horizontal extension for the application of URANOS (see RC Line 181). The air buffer layer is, as stated later, recommended to be constructed as a 1000 m thick column.

RC:   *Line 169. Please, adopt the term particle density which is more commonly used in the soil science literature over "compound density".*

AR:   Change made.

RC:   *Line 172. This is what I meant earlier about "further" moderating the detector.*

AR:   The sentence has been rephrased to

> In order to assess the effect the additional moderating effect of the directional shielding on the measured intensity...

RC:   *Line 181. The virtual detector is within a domain of 800 x 800 m, but the authors defined earlier that the domain was only 600 m. Please clarify.*

AR:   The paragraph "Neutron simulation model URANOS" is a general introduction to the URANOS Monte Carlo model. We have rephrased the sentence of concerning 'at least 600 m' there. The following paragraph "Scenarios simulated with URANOS" containing the specific number 800 m refers to the setting used in this study.

**RC:** *Line 194. Is this FOV of 180 degrees something that would be possible with an actual instrument? From figure 1 it seems more like 90 degrees.*

AR:  For clarification, we have added the sentence

> This value [of the FOV] has no direct relation to the geometry of the opening face of the shielding. Instead, it is an arbitrary decision to target the area within the FOV with the measurement.

**RC:** *Line 195.  In this section, are the authors referring to count rate based on raw neutron counts or on corrected counts by atmospheric conditions and incoming neutron flux?*

AR:  Any correction (incoming, barometric, water vapour) would linearly affect all mentioned rates the same way, so the given equations hold for either.

**RC:** *Line 201. Remove word "Please".*

AR:  Done.

**RC:** *Line 202.  Why not abbreviating gamma as "D" (directional) and "OD" (omnidirectional) or "uni" and "omni" for better readability across the manuscript.*

AR:  We changed the notation to $\gamma_D$ and $\gamma_{!D}$ for consistency with our notation. This also affected $R_s$ being changed to $R_D$.

**RC:** *Figure 4.  Please indicate in the caption the assumption that R1total < R2total (I assume this since the bars for total counts are different for R1 and R2). Although I'm confused since in Line 215 it says that there is no fundamental different between A1 and A2.*

AR:  Line 215 was ambiguous, we rephrased to

> The directional detector may be oriented towards the left half plane $A_1$ or the right half plane $A_2$. Formally, there is no fundamental difference in the equations for each and we exemplify the next step with the one oriented towards $A_1$ and its count rate $R_{f1,alb}$.

**RC:** *Lines 290-300. These sections probably need to be merged into a single section and presented as separate paragraphs. The "Count rate" section is only one sentence long.*

AR:  We changed all subsection headings to simple bold face font like this:

> **Fraction of non-epithermal counts** ($\epsilon$): In the examples, we illustrate ...

**RC:** *Line 319. How much higher is the reduction in count rate by adding the shield beta in wet conditions? Even a speculation would be fine here. Perhaps the authors can use soil saturated conditions as a reference. Do you need to add a reference to Table 3 in this statement?*

AR:  Actual values for $\beta$ can be abstracted from the table. We clarified this by modifying the sentence to:

> Secondly, the total count rate reduction by adding the shield, $\beta$, is at least 30 %. For the wetter conditions, it is even higher, reaching $\beta$=40 % for $\theta$=50 % (see **??**).

**RC:** *Table 3. What is the integration time of these neutron counts? Are they corrected for atmospheric conditions? Is the "no shield" term referring to a bare detector or to the portion of the directional detector with "less" shielding? Please clarify.*

 AR: All details given in section 3.1 result from the neutron simulation, with its details given in section 2.1. The integration times (i.e. total counts simulated) have been chosen to yield sufficiently robust statistics. Corrections would have no affect on the ratios reported here (see reply to RC Line 195). "No shield" refers to the omnidirectional detector, as now clarified in the text (see reply to RC Lines 126).

**RC:** *Line 354. I'm starting to think that it is hard to remember what "worstcase" scenario means at this point. The definition of this term occurred several sections above and the name does not seem intuitive. I wonder if there is a better way of naming.*

 AR: We had used the terms "pessimistic" before, but found this to sound too subjective. "favourable" and "unfavourable" could be an option, which we would implement.

**RC:** *Figure 9. Please specify for which sensor form factor this figure applies. Does this figure translate to other detector configurations?*

 AR: All computations refer to the sensor geometry described in Fig.1. This point is also addressed in the Discussion, but will be more prominent after the restructuring of this section (see RC "Limitations and Outlook")

**RC:** *Line 410. Larger planar shielding in what direction of the instrument. It will be good to provide a more explicit comment for other researchers that want to replicate or design their own directional detectors.*

 AR: The respective sentence was modified to

> Other designs (e.g. larger vertical planar shieldings blocking off one half space) may provide superior characteristics, namely $\eta$, which could be assessed by further neutron simulations.

**RC:** *Line 396. How does this integration time relate to typical applications for soil moisture sensing in agricultural and hydrological scenarios? It will be good for the authors to expand on the practical applications/obstacles of the required time integrations and counts required to achieve a certain precision.*

 AR: We are somewhat hesitant to claim general figures on what aggregation times are required, as they may be highly application-specific. However, we now refer to the respective explanations given before and modified the sentence to

> 1, 6, 12 and 24 h. Longer aggregation times are not recommended from a hydrological perspective, since they would commonly imply too high a change of the observed variable during that interval (e.g. due to rainfall or drying and respective change in $R$.

---

## Author Response (AR1)

**Revision 1: Author Response to Referee #1, #2, #3**

**Assessing the feasibility of a directional CRNS-sensor for estimating soil moisture**

Till Francke et al.
*Geoscientific Instrumentation Methods and Data Systems, Discuss.,* `doi:10.5194/gi-2021-18`
* * *
**RC:** *Reviewer Comment*,     AR: *Author Response*,     ☐ Manuscript text

Dear Referees, dear Editor,

we would like to thank you again for your comments and constructive suggestions to our manuscript. In parts, this letter contains our responses already given in the interactive discussion, which also formed the basis for the revision of the manuscript. It also contains the response to Reviewer 3, which was not part of the Discussion phase, as it had been closed directly after Reviewer 3's submission.

The revised version of the manuscript has substantially changed and, in our view, improved thanks to the given suggestions. We hope that this revised version meets the approval of the reviewers and the editor.

Kind regards,
Till Francke
(on behalf of the author team)

**1. Response to Referees #1 and #2**

**1.1. General comments**

**RC:** *Review of manuscript titled "Assessing the feasibility of a directional CRNS-sensor for estimating soil moisture". The idea is fantastic and very much needed.*

 AR: We very much appreciate the reviewer's positive feedback.

**RC:** *I think this is study that will gain the interest of the research community. The main limitations of the manuscript are:*

**1.2. Background information on CRNS**

- the assumption that readers are familiar with the cosmic-ray neutron sensing technology for measurement of soil moisture.

 AR: We indeed consider our manuscript to address readers familiar with the basic concept of CRNS, before embarking on the more intricate journey of directional sensing. However, we added some introductory information in this regard to section 1.1, which now reads

> CRNS relies on the measurement of ambient epithermal neutrons. The amount of hydrogen in the vicinity of the sensor governs how neutrons of this energy level are slowed-down in collision processes. Thus, the count rate of epithermal neutrons is inversely related to the hydrogen inventory and can be used to infer the abovementioned environmental variables. Consequently, a major advantage of the CRNS method is its non-invasive character, as opposed to traditional measurements of soil moisture as e.g. thermogravimetric or electromagnetic (FDR, TDT, TDR) methods. Additionally, the measured cosmic-ray neutrons naturally integrate over an area with approximately 150 m radius and a depth of typically 2-4 decimeters [Köhli et al., 2015], opposed to the traditional point-scale measurement. This results in practical and representative estimates of soil moisture at the field scale. This intermediate scale of measurement support effectively bridges the gap between traditional point measurements and coarser large-scale products from remote sensing or hydrological modelling.

**1.3. Prototype**

**RC:** *- The manuscript is dense and the logical arrangement of ideas together with some limitations in the English language makes this paper difficult to read. It starts by describing a previous directional sensor, but then it shifts the attention to simulation work, without interconnecting both. It seems to me that the manuscript could be much shorter and more straight to the point*

AR: We are confident that the reviewers' suggestions (below) helped to improve the legibility of the manuscript.

The description of the existing directional sensor is meant as a justification why we chose to simulate this very realization of a shielded instrument as an example case, and other designs are likely imaginable. Its first data set retrieved in the field is now shown exemplary (Fig. 1) and is actually very much the set-up being the starting point for the simulation work (Fig. 3). We have strengthened this reasoning and interconnected both parts. Details on the implemented changes are listed in the reply to Reviewer 3's comments (see Reply section 3.2), who rather suggested to expand on the details of the prototype, instead of shortening the manuscript. We hope to have found a reasonable compromise between both suggestions.

**1.4. Clarifying structure of manuscript**

**RC:** *At time it seems that this manuscript could have been partitioned in two different manuscripts parts: a more theoretical description of the error sources, propagation, and computation of directional neutron counts, and a second part about testing the quantitative reasoning with a modeling effort.*

AR: These two parts are actually already distinguished in the "Method" subsection (2.1 and 2.2.) and the corresponding "Results and Discussion" subsections (3.1. and 3.2): *.1 deals with the numerical modelling, necessary to understand the relevant effects and determine actual parameter values for describing them (e.g. parameter $\beta$), while *.2 generalizes these findings to a broader range of settings in terms of count rates, signal contrast and aggregation times using an analytical approach. We made this clearer in the manuscript by

- emphasizing this concept again in the beginning of the section 2.2:

> This analytical approach allows us to assess the feasibility of directional CRNS in a wide range of settings without the need to perform numerous computationally-demanding numerical simulations.

- guiding the reader with an additional introductory sentence in the beginning of the Results section:

> This section presents the results of the detailed numerical simulations (section 3.1), followed by their generalisation using the analytical approach (section 3.2).

- renaming section 3.1 from "Neutron simulation" to "Numerical neutron simulations"

- correcting the hierarchy of former sections "3.2 Angular sensitivity" and "3.3 shielding effect" to subsections of 3.1 "Numerical neutron simulations" (i.e. 3.1.2 and 3.1.3) to correspond to the structure used in the Methods section.

**1.5. Restructuring "Limitations and Outlook"**

**RC:** *The "Limitations and Outlook" section contains valuable information for the research community. I suggest condensing this section into a fewer, but still relevant, number of points, or merging the points into a narrative. With so many bullet points I question the usefulness of the current layout. Even this section could be interpreted as a potential technical note or manuscript presenting and discussing limitations and opportunities of directional detectors.*

AR: We agree that the previous layout was unfortunate. We restructured the section by grouping specific items in meaningful sub-topics, i.e. "Assumptions made in the modelling and choice of parameters", "Implications for practical application of a directional sensor" and "Further open questions for follow-up studies".

**1.6. Specific comments, details**

**RC:** *Line 14. Define again the abbreviation for Cosmic Ray Neutron Sensing (CRNS) in the manuscript narrative.*

AR: Done within the heading of first subsection: "Cosmic ray neutron sensing (CRNS) in environmental sciences"

**RC:** *Line 14-15: The first two sentences can probably be merged into one. Consider the following alternative: "Cosmic Ray Neutron Sensing (CRNS) has been widely adopted in the past decade to measure soil water content in environmental sciences" or this one "In the past decade, the adoption of Cosmic Ray Neutron Sensing (CRNS) has increased considerably to measure soil water content in hydrological, agricultural, and environmental research applications (Zreda et al., 2008)"*

AR: The second suggestion has been implemented.

**1.7. Clarification of field of application**

**RC:** *Line 15. Unclear statement "in both research and application". Research could be basic or applied, so its unclear whether the authors refer to applied research or applications beyond research (end user or consumer applications). Please clarify.*

AR: We modified the sentence to

> Such measurements could serve a variety of purposes in both, research and end user applications,...

**1.8. Editorial work**

**RC:** *Line 17. Remove the first "or" so that "..., to support irrigation management..."*

**RC:** *Line 22: Re-word to "a depth of a few decimeters" It may be good to provide a quantitative range. It seems that the typical sensing depth oscillates between 10 and 40 cm depending on the soil moisture conditions.*

**AR:** Suggestions above have been implemented.

**1.9. Omnidirectionality**

**RC:** *Line 25. This paragraph probably needs more context and depth. At this point it's not obvious that the CRNS is omnidirectional, so it may be good to state that this is the nature of current measurements and available instruments. I suggest briefly expanding more on the issue of instrument omnidirectionality (why is it this way?) and other factors such as soil spatial heterogeneity that can create soil moisture spatial patterns. I think this paragraph would be essential to understand the motivation of the study.*

**AR:** The paragraph now reads:

> However, the larger spatial support of the omni-directional measurement compared to the point-scale methods comes at the cost of spatial resolution: The neutron sensor registers neutrons having interacted with the soil in the so called "footprint radius". It does not discern the direction nor the distance of the point, where this interaction took place.

**1.10. Editorial work**

**RC:** *The "larger spatial support" compared to what measurement? point-level?*

**AR:** The comparison refers to classic point scale measurements. The sentence now reads:

> However, the larger spatial support of the omni-directional measurement compared to the point-scale methods ...

**1.11. Editorial work**

**RC:** *Line 30. Specify that the "energy level separation" is for the spectrum of cosmic-ray neutrons.*

Line 33. Consider a better transition here. For instance: "An approach to reconstruct sub-footprint patterns in soil moisture consists of using a dense. . . . (Heistermann et al., 2021). Although. . . "

Line 39. Consider merging this paragraph with the previous since it's a continuation of the same argument.

**AR:** –> All suggestions above have been implemented..

**1.12. Editorial work, cont'd**

**RC:** *Line 44. I would like to suggest the use of the following terms: "Directional neutron sensing" or "Cosmic-ray Directional Neutron Sensing"*

Line 48. There seems to be a typo here: "and/and"

Line 50. Consider replacing "defense" with military applications

Line 51. There is an extra ")"

AR: –> All suggestions above have been implemented.

**1.13. Restructuring and neutron energy levels**

**RC:** *Line 55. This paragraph seems too short at only two sentences long. I suggest detailing the type or energy level of the incoming neutrons. It's unclear from the text whether the authors are referring to thermalized or epithermal neutrons. It will also be important to highlight that the term "directional" in the context of this paper mostly applies to the horizontal plane.*

AR: The paragraph has been merged with its precursor. However, we do not think that in this context the energy level of the neutrons is of relevance and prefer to keep the message of the paragraph simple. A note on focusing on the *horizontal* aspect has been added to the end of section 1.2:

> Hence, the term "directional" in the following mostly applies to the horizontal plane, as implemented for the prototype presented.

**1.14. PE**

**RC:** *Line 59. Spell out "PE" (polyethylene?). Was this a high-density PE?*

AR: PE now spelled out. Mitrofanov et al. just report about the use of "polyethylene", leaving it open whether it was HDPE or not.

**1.15. Ntotal**

**RC:** *Line 61. How is Ntotal observed or computed? Is this the total considering the shielding or the total without it? I assume that for simultaneous collimation efficiency you need two devices or alternatively the same device with and without the shielding over short periods of time.*

AR: The description was ambiguous and modified to

> ...$\eta$ being the ratio between counts from the targeted FOV $N_{\text{FOV}}$ and the total counts $N_{\text{total}}$ registered by the detector, i.e. counts from any direction.

Thus, $N_{\text{total}}$ is the observable variable, which can directly be measured by the directional detector. No concomitant operation of an unshielded detector is required.

**1.16. Shielding vs moderator**

**RC:** *Line 70. Please, clarify whether the unshielded side means less shielding to avoid confusion with a bare detector. For instance, the CRS 2000B already includes some shielding to attenuate neutrons. This means that the authors added an extra layer of shielding.*

AR: Indeed, the sentence refers to the extra-layer as added in the form of the shielding. We added the sentence

> This shielding is independent of the moderator already in place around the CRS 2000B used to thermalize the neutrons before being detected.

Moreover, the clarifying definition of the terms ("moderator" vs. "shielding") in the introduction should help to resolve this ambiguity.

**1.17. Description of prototype**

**RC:** *Line 73. Replace 2Pi by 360 degress. What do the authors mean with "flexible"? What is the integration time at each angle? or is it a continuous scanning? I think that a few more details will allow readers to reconstruct the idea and reproduce the study.*

AR: Changes made and requested information added:

> This could be operated to cover the full $360°$ periphery ($2\pi$) in flexible angular sections with configurable integration times at selected positions, and thus allow for measurements with variable directions producing time-series of count rates for different FOV in the footprint.

**1.18. Bycatch**

**RC:** *Line 99. Would this bycatch flux be similar regardless of the gas used in the detector ($^3$He vs $^{10}$BF3)?*

AR: To our knowledge, the mentioned 'bycatch' depends only to a minor extent on the gas type, gas pressure and moderator configuration. Thus, the deviations are insignificant for the purpose of the discussion.

**1.19. Detector electronics**

**RC:** *Line 109. Emphasize that these settings are in the detector electronics. Can you add the model of the neutron pulse analyzer from Quaesta?*

AR: We changed the wording to "... threshold settings in the electronics " to underline the fact.

There are various forms of non-neutron signals. Some of them can be cut by setting lower and upper thresholds on the detected pulse energy. In the case of radioimpurities, their signatures are often indistinguishable from neutrons in case their energy lies in the range of those of neutron conversion products. With usual electronics, independent of the manufacturer, such cannot be discriminated against. They form a plateau 'under' the neutron events.

**1.19.1 Bare vs. unshielded**

**RC:** *Lines 126. The wording in this sentence implies that there is an "unshielded side", which most people will take as a bare detector, which I suspect is not the case here.*

AR: To avoid this potential confusion we have added the following sentence to the end of section 1.1.:

> The term "shielding" is often used to denote parts around the detector to moderate ("thermalize") high energy neutrons to be detectable. Here, we will use the term "moderator" for this component, while components aiming to achieve directionality are referred to as "shielding".

**1.20. Figure 2**

**RC:** *Figure 2. I like this figure. To make it a bit more explicit it will be good to add in the description the meaning of the shaded area. Are the arrows pointing in the "increasing" direction? Perhaps adding a "+" and "-" symbols could help the reader. Will it be possible to add magnitudes to the axes and denote with a point the selected combination for this study?*

**AR:** The shaded areas had no actual meaning apart from being not part of the inner triangle. To avoid this confusion, we have modified the figure, its caption and added "+" and "-".

[Figure]

Figure 1: "Tradeoff-triangle" in directional CRNS-measurements as ternary plot: for a given combination of two of these parameters, the third parameter must be adjusted accordingly to obtain the requested accuracy. The inner triangle (grey) illustrates the parameter space for an increase in required accuracy. The straight outline here is only chosen for the sake of simplicity and could be curved instead.

We intend this figure merely as a conceptual illustration of the trade-off problem. Thus, we'd rather refrain from adding actual numbers. Moreover, the performed multi-dimensional analysis of each of the three factors would result in multiple inner triangles, which would unnecessarily complicate the figure.

**1.21. Ambiguous use of "area"**

**RC:** *Line 129. Unclear whether the authors are referring to the area in the FOV (i.e. area of the open side of the instrument) or the horizontal area part of the sensing footprint.*

**AR:** We have added a reference to Figure 3 to make clear that "area" refers to the part of the footprint. The sentence now reads:

> Directional neutron sensing aims to determine the neutron flux rate $R_1$ that is characteristic of the area $A_1$ in the field of view (see Fig. 3).

**1.22. Feasibility of in-situ experiments**

**RC:** *Line 141. While in situ experiments controlling the neutron flux are impractical, in situ validation of the approach is not. For instance, a field validation could be done by intensively measuring soil moisture. For this to work a field with a known directional variability may need to be selected. I just wanted to emphasize that in situ validation may not impractical (in terms of measuring soil moisture).*

**AR:** We agree that controlling the flux rates in two half-spaces via adjusting the soil moisture is, in theory, conceivable. And we now have included an example data set from a quite extreme case, that is a water body half-space vs. a land surface half-space, though not being ideal in contrast as the soil was wet during this period (Fig. 1). However, judging from previous efforts to reliably measure (let alone adjust and hold) the soil moisture to target values in sufficiently large test sites, creating such settings is a very difficult endeavor. Thus, we replaced the term "impracticable" with "difficult to implement".

**1.23. Term "scanning"**

**RC:** *Line 149. I like the first question, but it may need to be formulated differently. As it stands now it will be more relative to the ability to regulate the rotary mechanism than the detector itself.*

**AR:** We posed the question in the most general way. As the results later will show, the required aggregation times are in the range of hours, so the velocity of the rotary mechanism is by no means a limitation. However, to avoid the confusion, we have replaced the term "scanning" (implying continuous rotation) by "directional measurement" (measurement with constant orientation over arbitrarily long periods) throughout the manuscript.

**1.24. Description of contrast**

**RC:** *Line 150. The authors vaguely described what is meant with "contrast". Perhaps a bit more background on this metric would be helpful. Most people are familiar with count rates and integration time, perhaps less familiar with signal contrast (unless the authors are referring to signal-to-noise ratio)*

**AR:** The description of "contrast" three paragraphs above has been extended and now reads

> The signal contrast between the two, i.e. the relative difference in the two flux rates ($\Delta R$, see Eq. 19) is eventually determined by the different hydrogen inventories in the two areas.

**1.25. Metrics**

**RC:** *Line 151. What is a robust estimate? It would be nice to provide a more objective metric, perhaps in terms of error in volumetric water content (or the equivalent neutron count).*

**AR:** We deliberately used the general expression "robust metric" here, as robustness can be related either to *distinguish* or to *determine* rates, as explained in the sentence following question C.

We also strictly limit the analysis to the count rates, not soil moisture, for the reasons stated in the "Limitations"

section:

> This study exclusively considered count rates as the target observation variable. However, for application, count rates are merely a proxy which need to be converted to the actual quantity of interest, namely soil moisture, snow or biomass. The relationships used in such conversion (e.g. [Desilets et al., 2010]) are not linear; instead, they saturate with increasing hydrogen pools and low count rates. [...] Combining the presented approach with the one of [Jakobi et al., 2020] would allow the direct quantification of uncertainty for the target variable.

**1.26. Extent model domain**

**RC:** *Line 165. Is the stretch of 600 m in all directions? or just in the horizontal direction?*

**AR:** 600 m is meant as the minimum horizontal extension for the application of URANOS (also see Reply section 1.29). To clarify this, we modified the sentence to

> The URANOS model can use setups with either open domains of at least 600 m or smaller sizes with periodic or reflecting boundary conditions.

The air buffer layer is, as stated later, recommended to be constructed as a 1000 m thick column.

**1.27. Terminology**

**RC:** *Line 169. Please, adopt the term particle density which is more commonly used in the soil science literature over "compound density".*

**AR:** Change made.

**1.28. Moderator**

**RC:** *Line 172. This is what I meant earlier about "further" moderating the detector.*

**AR:** The sentence has been rephrased to

> In order to assess the effect the additional moderating effect of the directional shielding on the measured intensity...

**1.29. Extent model domain, cont'd**

**RC:** *Line 181. The virtual detector is within a domain of 800 x 800 m, but the authors defined earlier that the domain was only 600 m. Please clarify.*

**AR:** The paragraph "Neutron simulation model URANOS" is a general introduction to the URANOS Monte Carlo model. We have rephrased the sentence of concern to 'at least 600 m' there. The following paragraph "Scenarios simulated with URANOS" containing the specific number 800 m refers to the specific setting used in this study, which should be clearer now.

**1.30. Incoming correction**

**RC:** *Line 194. Is this FOV of 180 degrees something that would be possible with an actual instrument? From figure 1 it seems more like 90 degrees.*

**AR:** For clarification, we have added the sentence

> This value [of the FOV] has no direct relation to the geometry of the opening face of the shielding. Instead, it is an arbitrary decision to target the area within the FOV with the measurement.

**1.31. Incoming correction**

**RC:** *Line 195. In this section, are the authors referring to count rate based on raw neutron counts or on corrected counts by atmospheric conditions and incoming neutron flux?*

**AR:** Any correction (incoming, barometric, water vapour) would linearly affect all mentioned rates the same way, so the given equations hold for either.

**1.32. Style**

**RC:** *Line 201. Remove word "Please".*

**AR:** Done.

**1.33. Notation Gamma**

**RC:** *Line 202. Why not abbreviating gamma as "D" (directional) and "OD" (omnidirectional) or "uni" and "omni" for better readability across the manuscript.*

**AR:** We changed the notation to $\gamma_\mathrm{D}$ and $\gamma_{!\mathrm{D}}$ for consistency with our notation. This also affected $R_\mathrm{s}$ being changed to $R_\mathrm{D}$.

**1.34. Figure 4**

**RC:** *Figure 4. Please indicate in the caption the assumption that R1total < R2total (I assume this since the bars for total counts are different for R1 and R2). Although I'm confused since in Line 215 it says that there is no fundamental different between A1 and A2.*

**AR:** Line 215 was ambiguous, we rephrased to

> The directional detector may be oriented towards the left half plane $A_1$ or the right half plane $A_2$. Formally, there is no fundamental difference in the equations for each and we exemplify the next step with the one oriented towards $A_1$ and its count rate $R_{\mathrm{f1,alb}}$.

**1.35. Merging small sections**

**RC:** *Lines 290-300. These sections probably need to be merged into a single section and presented as separate paragraphs. The "Count rate" section is only one sentence long.*

**AR:** We changed all subsection headings to simple bold face font like this:

> **Fraction of non-epithermal counts** ($\epsilon$): In the examples, we illustrate ...

**1.36. Table 3**

**RC:** *Line 319. How much higher is the reduction in count rate by adding the shield beta in wet conditions? Even a speculation would be fine here. Perhaps the authors can use soil saturated conditions as a reference. Do you need to add a reference to Table 3 in this statement?*

**AR:** Actual values for $\beta$ can be retrieved from the table. We clarified this by modifying the sentence to:

> Secondly, the total count rate reduction by adding the shield, $\beta$, is at least 30 %. For the wetter conditions, it is even higher, reaching $\beta$=40 % for $\theta$=50 % (see Table 4).

Please note that numbering of the respective table changed to "Table 4" (see Reply section 3.3).

**1.37. Table 3, cont'd**

**RC:** *Table 3. What is the integration time of these neutron counts? Are they corrected for atmospheric conditions? Is the "no shield" term referring to a bare detector or to the portion of the directional detector with "less" shielding? Please clarify.*

**AR:** All details given in section 3.1 result from the neutron simulation, with its details given in section 2.1. The integration times (i.e. total counts simulated) have been chosen to yield sufficiently robust statistics. Corrections would have no affect on the ratios reported here (see Reply section 1.31). "No shield" refers to the omnidirectional detector, as now clarified in the text (see Reply section 1.19.1). Please note that the numbering of the respective table changed to "Table 4" (see Reply section 3.3).

**1.38. Wording "bestcase"/"worstcase"**

**RC:** *Line 354. I'm starting to think that it is hard to remember what "worstcase" scenario means at this point. The definition of this term occurred several sections above and the name does not seem intuitive. I wonder if there is a better way of naming.*

**AR:** We had used the terms "optimistic"/"pessimistic" before, but found this to sound too subjective. We now replaced "bestcase" and "worstcase" with "favourable" and "unfavourable", hoping that this is more intuitive. Figures and code have been updated accordingly.

We have also added a table summarizing the two scenarios (now Table 2, see Reply section 3.3).

**1.39. Fig. 9**

**RC:** *Figure 9. Please specify for which sensor form factor this figure applies. Does this figure translate to other detector configurations?*

**AR:** All computations refer to the sensor geometry described in Fig.1. This point is also addressed in the Discussion, and is now more prominent after the restructuring of this section (see Reply section 1.5).

**1.40. Planar shieldings**

**RC:** *Line 410. Larger planar shielding in what direction of the instrument. It will be good to provide a more explicit comment for other researchers that want to replicate or design their own directional detectors.*

**AR:** The respective sentence was modified to

> Other designs (e.g. larger vertical planar shieldings blocking off one half space) may provide superior characteristics, namely $\eta$, which could be assessed by further neutron simulations.

**1.41. Counts vs. theta**

**RC:** *Line 396. How does this integration time relate to typical applications for soil moisture sensing in agricultural and hydrological scenarios? It will be good for the authors to expand on the practical applications/obstacles of the required time integrations and counts required to achieve a certain precision.*

**AR:** We are somewhat hesitant to claim general figures on what aggregation times are required, as they may be highly application-specific. Moreover, the relationship between counts and theta depends on sensor-sensitivity and site characteristics, so there is no unique answer to this. However, we now refer to the respective explanations given before and modified the sentence to

> ... 1, 6, 12 and 24 h. Longer aggregation times are not recommended from a hydrological perspective, since they would commonly imply too high a change of the observed variable during that interval (e.g. due to rainfall or drying and respective change in $R$.

**2. Response to Referee #3**

**3.1. General comments**

**RC:** *The Authors present an interesting and valuable study on the feasibility of directional cosmic-ray neutron sensing (dCRNS). This scanning method could help the interpretation of the signal at specific sites with e.g., land use patches and it could provide some new opportunities for integrating the method for new applications. The manuscript is generally well written and clear and the analysis is based on several numerical simulations. For these reasons I think the study can be a valuable scientific contribution. I have however three main concerns that I think should be considered: [1] to add the experimental results; [2] to improve the quality of story-telling and [3] to strengthen the scientific value of the study with a more clear outlook. Despite these might require some additional work, I believe that the Authors will be able to well address my concerns. Some few specific comments are listed below that I hope can also help the overall improvements of the manuscript.*

**AR:** We appreciate Reviewer 3's positive feedback and have addressed the raised issues as follows.

**3.2. General comments [1]**

**RC:** *At the beginning of the manuscript a direction CRNS sensor has been introduced (fig1). Simulations are based on this sensor design. Surprisingly for me, no experimental results has been shown and discussed in the present study. Why that? I believe that some data have been collected since it is stated that this*

***prototype has been constructed since 2018 (L69). I think adding experimental results would strengthen the value of this study.***

AR: Reviewer 3 is correct assuming that the prototype has been used to collect some data. The first trial merely served as a proof-of-concept at the shore of a lake. No concomitant measurements on the terrestrial site are available for validation. Nevertheless, we have added an illustration of this trial to Figure 1:

[Figure]

Figure 2: CRNS sensor prototype with directional shielding, operated with alternating orientation towards a lake and land. Topleft: horizontal cross section; Topright: Installation with directional control; here the unshielded direction is oriented towards the viewer, i.e. the land site. The white box at the right contains the logger and modem. A conventional bare counting tube is mounted to the left. Bottom: Pilot measurement at a shore with alternating orientation towards the lake vs. land area, as half planes (cf. Fig. 3). Dots represent values for 30-min aggregation.

In the second trial, the experimental site did not feature half-planes with sufficiently distinct contrast in the hydrogen-inventory and, thus, count rates of the two half-planes. Additionally, the rotating mechanisms did not perform sufficiently stable, resulting in in an angular drift of the targeted directions. Not surprisingly,

the corresponding time series thus do not show different count rates for the half-planes. We have added this explanation to the first paragraph of section 3.5:

> So far, we have been unable to operate the prototype in a setting with sufficiently high contrast in count rates to practically demonstrate successful discrimination.

Thus, the prototype constituted the motivation for our theoretical study, not its subject. As the study shows, its actual application cannot be expected to yield distinguishable results in any but very high-contrasting and homogeneous settings. So far, we have not operated the sensor in these and cannot include useful data.

**3.3. General comments [2]**

**RC:** *[2] The manuscript well covers and discusses the methodology, the results and the limitations of the study. Despite I do not see any critical missing parts, I found the story line not always clear or a bit difficult to read. I think the manuscript could be improved by re-organizing especially the method section in a more synthetic structure that can be appreciated and understood even without reading all the details. A table summarizing the different scenarios simulated could also help the readings.*

**AR:** In fact, there only two scenarios "bestcase" and "worstcase" (now renamed to "favourable"/"unfavourable", see Reply section 1.38). To clarify this, we added a general explanation to the beginning of the Methods section 2:

> This assessment exploits two extreme scenarios termed "favourable" and "unfavourable", encompassing the most favourable and most adverse settings, respectively.

and in the beginning of section 2.2.6

> The following parameters where used to define two scenarios termed "favourable" and "unfavourable", encompassing the most favourable and most adverse settings, respectively:

Further addressing this issue, we changed the potentially confusing heading of section 2.1. from

> Scenarios simulated with URANOS

to

> Setups simulated with URANOS

Furthermore, we rearranged Section 2.2.6 and added the requested table summarizing the scenarios (Table 1 in this letter, now Table 2 in the revised manuscript). We also made various other changes to guide the reader through the story line (see reply sections 3.11 and 3.13).

Table 1: Parameter values describing the "favourable" and "unfavourable" scenarios

| Parameter | value in "favourable" | value in "unfavourable" |
|---|---|---|
| **Directional contribution** ($\eta$) | 0.72 | 0.61 |
| **Fraction of non-albedo neutrons in the directional detector** ($\gamma_\mathrm{s}$) | 0.2 | 0.31 |
| **Overall reduction of count rate due to the directional shielding** ($\beta$) | 0.4 | 0.3 |

**3.4. General comments [3]**

**RC:** *The final result seems to be that scanning CRNS is hardly feasible and only strong soil moisture changes and patches could be detected with long integration time. Despite I believe that scientific papers should also present and discuss "what does not work" to provide guidelines for further studies, I usually expect that from lab-field studies where it is hardly possible to repeat or to increase the numbers of experiments. In contrast, all the results of the present study are based on simulations. The Authors well discuss some limitations on the numerical settings and they often acknowledge that further simulations could shed lights on how to design an effecting settings (e.g., L411, L428, L438). The Authors conclude for instance that progress in detector technology and optimizing the shielding towards wider FOVs but more specificity could alleviate some of the restrictions and make directional scanning a useful tool for tailored use of CRNS (L501-502). I believe that the scientific value of the present study should be improved not speculating on that. Instead few specific additional simulations could support the design of a proper scanning CRNS in the future. Specifically, I personally suggest to consider a hypothetical very high sensitive sensor with a cone-shaped shielding (smaller angles) that has been shown in literature to be able to discriminate directional neutron fluxes but not in the context of soil moisture observations (see Becchetti et al, 2015). Alternative (or additional) settings based on the experiences of the Authors could be also considered. Despite I agree (see your comments at L442) that these set-ups could be not very practical nowadays considering among others the need of high sensitive (and probably expensive) sensors, it could be probably feasible in the near future considering current sensors developments. At such, the present study would provide a strong bases for further sensor development. At the present status, this is not the case.*

AR: Without doubt, it would be of high use to gain precise indications on promising new sensor developments (or, more precisely, the design of directional shielding). However, we would like to point out that the focus of our study is presenting a) an easy-to-build example of a directional shielding that can be rotated, b) specific computationally-expensive numerical simulations of this prototype and c) an analytical method to generalize the simulations to a wider range of settings to assess the general feasibility of directional CRNS.

We do not claim to exhaustively explore other designs for shieldings, which would require tremendously more neutron simulations and fundamental decisions on what can still be considered feasible, e.g. in terms of automatic rotation. However, regardless of the shielding design, the fundamental problem persists: The non-geometric transport of neutrons, which dominates over larger distances, poses fundamental restrictions on any directional neutron measurements in natural settings. This has been demonstrated with the "favourable" detector, showing that even with perfectly impermeable shielding material, directional specificity as expressed by directional contribution $\eta$, will not exceed 0.72 (see Table 5). To make this clear, we have added the following sentence to the conclusions:

This fundamental limitation is inherent to measuring neutrons under environmental conditions and cannot be alleviated by detector design and influenced only marginally by design of the shielding.

To further strengthen this point, please consider the following figure:

[Figure]

Figure 3: Histogram of simulated neutron origin angles for a hypothetical ideally-shielded detector registering only neutrons with incidence angles between 90 and 270 degrees.

This image illustrates the directional sensitivity of a perfect $180°$ shielding ($FOV_{shielding} = FOV_{targeted} = 180°$). As with the simulated prototype, the directional specificity decreases with distance. This again underlines that the main limitation lies in the non-geometric transport of the neutrons over longer distances – a characteristic that cannot be influenced by the shielding. For this hypothetical "ideal" shield, the directional contribution is $\eta \approx 0.725$. Likewise, this value cannot be exceeded by any shield blocking $180°$. However, higher values might, at least theoretically, be obtained when the "constructional FOV ($FOV_{shielding}$) is further narrowed, while keeping the area of interest ($FOV_{targeted}$) constant. As this comes at the cost of a reduction in total counts, it again would require an extensive analysis to find an optimal balance between $\eta$ and count rate such that the signal of the different viewing angles can be statistically distinguished and determined.

**3.5. Specific comments [1]**

**3.6. Rover shielding**

**RC:** *L66. As far as I have understood, side-shielding has also been suggested for addressing neutrons form below-sensor in the so-called "road effect" by Schrön et al. 2018? If this is the case it is worth extending*

*here the discussion with this example and clarifying that directional shielding has been presented for discriminating neutron fluxes in the vertical directions but so far not focusing on lateral side-directions. Schrön, M., R. Rosolem, M. Köhli, L. Piussi, I. Schröter, J. Iwema, S. Kögler, et al. "Cosmic-Ray Neutron Rover Surveys of Field Soil Moisture and the Influence of Roads." Water Resources Research, May 14, 2018. https://doi.org/10.1029/2017WR021719.*

AR:   Following this suggestion we have added the reference to section 1.2, explaining that we do not consider these bottom shields used in roving to be directional CRNS in the narrower sense:

> Conversely, Schroen et al., 2018 and Schroen et al. 2021 have used a shielding to block neutrons from directly below a CRNS rover unit to reduce the so-called "road effect". Although this might also be considered as a case of directional CRNS in the wider sense, its main aim is rather excluding a specific direction than focusing on one.

**3.7.   Relocating description of prototype**

RC:   *L69-80. These descriptions could better fit the methodology section.*

AR:   This paragraph contains the description of the prototype directional sensor. As laid out in the reply to General comment [1] (see Reply section 3.2), the actual prototype provided rather the motivation to the study, but does not serve as a method. Listing its description in the Methods section would imply that any actual measurements would be the subject of the study. This is not the case, as we have clarified with the changes mentioned in the reply to General comment [1] (see Reply section 3.2).

**3.8.   Relocating section 1.3**

RC:   *Section 1.3 fits better the methodology section, in my opinion.*

AR:   We agree that parts of section could also be placed in the methodology section. However, section 1.3 introduces terminology, symbols and concepts (e.g. the origins of the uncertainty) that are necessary for laying out the Specific challenges in section 1.4, concisely explaining the concept of the tradeoff-triangle and formulating the objectives. We therefore suggest that the current position of section 1.3. enhances legibility.

**3.9.   General feasibility of directional CRNS**

RC:   *L117-121. Based on this explanation, I would have expected that directional shielding is not feasible.*

AR:   This explanation indeed contains the inherent limitation of directional neutron measurement in environmental conditions. However, we have rephrased it to avoid the impression of unfeasibility:

> This phenomenon is especially pronounced for origins far from the sensor. Consequently, the incidence angle of a neutron reaching the detector is less correlated with its angle of origin, the farther the distance between origin and detector.

**3.10.   Unclear sentence structure**

RC:   *L179. Remove "it"*

AR:   The respective sentence was rephrased to

> This can be regarded as a suitable approximation for the purpose of investigating the distance-dependent angular response.

**3.11. Epithermals**

**RC:** *L182. There is a quite long discussion on non-epithermal neutrons. Most of this contribution, as far as I have understood, comes from thermal energy neutrons (20%). But it is then stated the Thermal neutron transport was disabled for reasons of computational speed. I'm a bit confused by the importance of this discussion and the detailed description at L101-113.*

**AR:** The sentence was modified to

> Thermal neutron transport was disabled for reasons of computational speed (please note that their effect on signal noise is still included in the generalisation of the simulation, section 2.2)

**3.12. Typo**

**RC:** *L186. Remove one "the"*

**AR:** Done.

**3.13. (Sub-)section titles**

**RC:** *L310. As far as I have understood, currently all are neutron simulations. Why then the need of the title section 3.1?*

**AR:** In a wider sense, all included steps could be considered "neutron simulations". However, our approach consists of the computationally demanding numerical neutron simulations with URANOS (actual "Neutron simulations") used to infer the basic parameters, and the generalisation of these findings using analytical expressions ("Generalisation"). We made this clearer in the manuscript by

- emphasizing this concept again in the beginning of the section 2.2:

> This analytical approach allows us to assess the feasibility of directional CRNS in a wide range of settings without the need to perform numerous computationally-demanding numerical simulations.

- guiding the reader with an additional introductory sentence to the Results section:

> This section presents the results of the detailed numerical simulations (section 3.1), followed by their generalisation using the analytical approach (section 3.2).

- renaming section 3.1 from "Neutron simulation" to "Numerical neutron simulations"

- correcting the hierarchy of former sections "3.2 Angular sensitivity" and "3.3 shielding effect" to subsections of 3.1 "Numerical neutron simulations" (i.e. 3.1.2 and 3.1.3) to correspond to the structure used in the Methods section

**3.14. Abbreviation in conclusion**

**RC:** *L487. I suggest specifying full name in the conclusions.*

AR: Done.

**3.15. Designing optimized shieldings**

**RC:** *L501. I strongly encourage the Authors to provide more insights on how optimizing the shielding (see general comment [3]).*

AR: Please see reply to "General comment [3]" above (Reply section 3.4).

**References**

[Desilets et al., 2010] Desilets, D., Zreda, M., and Ferré, T. P. A. (2010). Nature's neutron probe: Land surface hydrology at an elusive scale with cosmic rays. *Water Resources Research*, 46:W11505.

[Jakobi et al., 2020] Jakobi, J., Huisman, J. A., Schrön, M., Fiedler, J., Brogi, C., Vereecken, H., and Bogena, H. R. (2020). Error estimation for soil moisture measurements with cosmic ray neutron sensing and implications for rover surveys. *Frontiers in Water*, 2:10.

[Köhli et al., 2015] Köhli, M., Schrön, M., Zreda, M., Schmidt, U., Dietrich, P., and Zacharias, S. (2015). Footprint characteristics revised for field-scale soil moisture monitoring with cosmic-ray neutrons. *Water Resources Research*, 51(7):5772–5790.